# Model-Preserving Adaptive Rounding

**Albert Tseng** [1]  **Zhaofeng Sun** [1]  **Christopher De Sa** [1]

## Abstract

The goal of LLM quantization is to produce a compressed model whose output distribution is as close to the original model's as possible. To do this tractably, most quantization algorithms minimize the immediate activation error of each layer as a proxy for the end-to-end error. However, this ignores the effect of future layers, making it a poor proxy. In this work, we introduce Yet Another Quantization Algorithm (YAQA), a new adaptive rounding algorithm that directly considers the error at the network's output. YAQA introduces a series of theoretical results that culminate in the first end-to-end error bounds for quantization algorithms. First, we characterize the convergence time of adaptive rounding algorithms via the structure of their Hessian approximations. We then show that the end-to-end error can be bounded by the approximation's cosine similarity to the true Hessian. This admits a natural Kronecker-factored approximation with corresponding near-optimal Hessian sketches. YAQA is provably better than GPTQ/LDLQ and empirically reduces the KL divergence by $\approx 30\%$ over these methods. YAQA even achieves a lower KL than quantization aware training. This translates to state of the art performance on downstream tasks, all while adding no inference overhead.

## 1. Introduction

Modern large language models (LLMs) contain billions of parameters, posing challenges for efficient deployment (Team et al., 2025; Tseng & De Sa, 2026). One way to improve the cost-benefit tradeoffs of LLM inference is by quantizing high precision model parameters $\theta^*$ to low precision representations. In memory-bound settings, this reduces bandwidth, and in compute-bound settings, hardware-

supported datatypes increase throughput. At their core, quantization frameworks (e.g. QuIP#) consist of two components: a quantizer $\mathcal{Q}$ (e.g. E8P) that defines a set of representable points $C$, and a rounding algorithm (e.g. LDLQ) that assigns $\theta^* \to C$. While quantizers are restricted by inference constraints, rounding algorithms give flexibility in the final model.

The goal of a rounding algorithm is to best preserve the original model outputs:

$$\hat{\theta} \leftarrow \underset{\theta \in C}{\arg\min} \ \ \mathbb{E}_{X \sim \mathcal{D}} \left[ D_{\mathbf{KL}}(M(\theta^*, X) \| M(\theta, X)) \right],$$
(1)

where $M$ is a model architecture, $M(\theta^*, X)$ are the original model outputs, $M(\theta, X)$ are the quantized model outputs, and $X$ is a input example sampled from a data distribution $\mathcal{D}$. Exactly solving Equation 1 is intractable due to the combinatorial nature of discrete optimization. Instead, modern quantization frameworks either optimize it with first-order descent-style methods such as quantization aware training (QAT), or second-order adaptive rounding algorithms such as GPTQ (Team et al., 2025; Frantar et al., 2023). The former requires significant amounts of compute to "learn" quantized representations, and the latter algorithmically rounds individual layers with second-order statistics.

To round a layer with second-order statistics, let $\mathcal{L}(W)$ denote the loss of Equation 1 resulting from changing that layer's weights to $W$, leaving all other weights alone with their value in $\theta^*$. Then,

$$\mathcal{L}(W) \approx \frac{1}{2} \text{vec}(W - W^*)^T (\nabla^2 \mathcal{L}(W^*)) \text{vec}(W - W^*),$$
(2)

where $W^* \in \mathbb{R}^{m \times n}$ are the original weights and first order terms involving $\nabla \mathcal{L}(W^*)$ are 0 since $D_{\mathbf{KL}}$ is minimized at the original model. Note that this is just a standard second order approximation of the end-to-end loss for a single layer. Since $\nabla^2 \mathcal{L}(W^*) \in \mathbb{R}^{mn \times mn}$ is too large to directly operate on, modern adaptive rounding algorithms use information from structured approximations (colloquially "Hessians") of $\nabla^2 \mathcal{L}(W^*)$ to round elements of $W^*$.

For example, the widely used GPTQ algorithm rounds using the Hessian of the immediate layerwise activation error as a proxy for $\nabla^2 \mathcal{L}(W^*)$ (Frantar et al., 2023). Like with $\mathcal{L}$, the minimum of this objective is achieved at the original model,

[1]Department of Computer Science, Cornell University. Correspondence to: Albert Tseng <albert@cs.cornell.edu>.

*Proceedings of the 43rd International Conference on Machine Learning*, Seoul, South Korea. PMLR 306, 2026. Copyright 2026 by the author(s).

but since this ignores the effect of quantization on future layers, it is not a very good proxy. Newer methods such as GuidedQuant and SqueezeLLM have explored using block diagonal approximations of the empirical Fisher to incorporate additional information from $\nabla^2 \mathcal{L}(W^*)$ (Kim et al., 2025; 2023). Unfortunately, these structured approximations are largely ad-hoc, with little theoretical justification for either the structure or approximation used. Resultingly, they underperform YAQA.

This begs the question: *what fundamental properties of Hessian sketches admit tractable adaptive rounding algorithms that produce high quality models?* In this work, we answer with Yet Another Quantization Algorithm (YAQA), an new adaptive rounding algorithm targeted for weight-only LLM post-training quantization that, *for the first time ever for any quantization algorithm, comes with bounds on the end-to-end quantization error.* YAQA generalizes LDLQ to consider the full model error, and is provably better at minimizing the KL while having the same asymptotic cost.

YAQA relies on a few key theoretical insights. We first characterize the convergence time of LDLQ and show that for it to be tractable, the Hessian approximation must have certain structural properties. This gives rise to a natural Kronecker-factored Hessian approximation that admits symmetric input and output-side feedback during rounding. Then, we show that the KL of YAQA's adaptive rounding algorithm can be bounded by the cosine similarity between its Hessian and $\nabla^2 \mathcal{L}(W^*)$. This motivates YAQA's Hessian sketches, which power iterate on $\nabla^2 \mathcal{L}(W^*)$ to obtain near-optimal Hessians. Empirically, YAQA reduces the KL by $\approx 30\%$ over LDLQ, achieves a lower KL than QAT, and sets a new state of the art in PTQ quality. In summary, we:

1. Introduce YAQA, a new quantization algorithm that generalizes local adaptive rounding methods to optimize the end-to-end KL and is provably better than GPTQ and LDLQ.

2. Characterize desiderata for Hessian sketches that admit tractable adaptive rounding algorithms and high quality quantized models, and prove the first ever theoretical bounds on the end-to-end quantization error.

3. Show that YAQA achieves a significantly lower KL than existing rounding algorithms, including QAT, and achieves state of the art downstream results.

## 2. Background

### 2.1. Large Language Model Quantization

LLM quantization approaches largely fall into two categories: post-training quantization (PTQ) and quantization aware training (QAT). In PTQ, models are compressed after training, whereas QAT and low-precision training methods produce *natively quantized models* through modified training recipes (Nagel et al., 2022; Tseng et al., 2025). Although PTQ has achieved popularity due to its combination of strong quality and relatively low cost, QAT can produce better models at the expense of needing significantly more compute. However, PTQ and QAT are not completely orthogonal. Recently, a class of PTQ approaches has emerged where weights are "learned" over a restricted subspace. For example, DiscQuant performs descent on the full model KL but limits parameters to either be rounded up or down, and PV-Tuning only optimizes a select subset of parameters at any given step (Chee et al., 2025; Malinovskii et al., 2024).

#### 2.1.1. LOCAL LAYERWISE ROUNDING

Most quantization algorithms use $H_1$, the Hessian of the immediate activation error ($\|x(W - W^*)^T\|_F^2$), as an approximation of $\nabla^2 \mathcal{L}(W^*)$. $H_1 \in \mathbb{R}^{n \times n} = \mathbb{E}[x^T x]$, where $x \in \mathbb{R}^{1 \times n}$ is the input activation to that layer and the expectation is taken over some data distribution. $H_1$ can be computed with only forward passes, making it cheap to obtain. In GPTQ and LDLQ, which are equivalent, columns in $W^*$ are iteratively rounded with linear feedback from the Cholesky decomposition of $H_1$. For the rest of this paper, we use LDLQ's presentation from (Chee et al., 2023) to refer to both. In AWQ, input channels are weighted with the diagonal of $H_1$ (Lin et al., 2023). Chee et al. (2023) showed that the error of LDLQ could be bounded by an increasing function of $H_1$'s incoherence, which intuitively measures how uniform important rounding directions are.

**Definition 2.1** ((Chee et al., 2023)). A Hessian $H \in \mathbb{R}^{n \times n}$ is $\mu$-incoherent if its eigendecomposition $H = Q\Lambda Q^T$ has $\max_{i,j} |Q_{ij}| = \max_{i,j} |e_i^T Q e_j| \leq \mu/\sqrt{n}$.

To reduce $\mu$ for $H_1$, Chee et al. (2023) introduced *incoherence processing*, which concentrates $H_1$ and $W^*$ by fast random orthogonal transformations. In QuIP#, Tseng et al. (2024a) did this with the randomized Hadamard transform (RHT), which also makes $W$ approximately Gaussian.

#### 2.1.2. FULL MODEL ADAPTIVE ROUNDING

Since $H_1$ ignores the effect of future layers on quantization, quantizing with $H_1$ does not always reduce the KL. Methods such as OBS, AdaRound, SqueezeLLM, and GuidedQuant have explored using additional information from the *empirical* Fisher matrix on the task loss (next-token cross entropy), which is fundamentally different from $\nabla^2 \mathcal{L}(W^*)$ (Hassibi et al., 1993; Nagel et al., 2020; Kim et al., 2023; 2025; Kunstner et al., 2020). These methods assume the model is trained to convergence, which is almost never true for LLMs (Hoffmann et al., 2022), and use block diagonal approximations to maintain tractability. Although these methods can outperform LDLQ, they do not generally come with error

bounds, making it difficult to reason about their characteristics. Furthermore, their Hessian sketches are largely ad-hoc, making it unclear if the resulting approximation is good or not. For example, increasing the fidelity of GuidedQuant's Hessian approximation does not consistently reduce the error. In contrast, as we show in Section 3.1, YAQA's KL is bounded by the cosine similarity between its Hessian and $\nabla^2 \mathcal{L}(W^*)$, which YAQA optimizes by construction.

### 2.1.3. GRADIENT-DESCENT-BASED METHODS

In all the aforementioned methods, quantized representations are obtained without "learning" and are not updated once they are obtained. In contrast, certain algorithms perform what is essentially constrained QAT with a much smaller compute budget. In PV-Tuning, learnable codebooks and code assignments are jointly optimized to minimize the end-to-end KL (Malinovskii et al., 2024). In DiscQuant, model weights are updated with gradient descent in a constrained subspace (Chee et al., 2025). In CBQ, quantized weights are learned with LoRA adapters and regularization in an AdaRound-style setup (Ding et al., 2025). In general, these methods are restricted to suboptimal scalar or unstructured vector quantizers. Finally, although some of these methods are guaranteed to perform local descent, QAT in general is not a descent method.

### 2.2. Hessian Approximations

While quantization works have mostly used $H_1$ or block diagonal Hessian approximations, prior *optimization* works have proposed a wide variety of Kronecker-factored Hessian sketches ($H \approx H_O \otimes H_I$) in the context of learning algorithms. Most sketches are based off the Fisher Information Matrix $\mathbb{E}[\text{vec}(\nabla_{W^*}\ell)\text{vec}(\nabla_{W^*}\ell)^T]$. In the real Fisher, which is equal to $\nabla^2 \mathcal{L}(W^*)$ (Gourieroux & Monfort, 1995), $\ell$ is computed with a Monte-Carlo sample over the model output logits. In the empirical Fisher, $\ell$ is the next token cross entropy. In KFAC, the authors show that for linear layers in MLPs, $H = \mathbb{E}[x^T x \otimes (\nabla_y \ell)^T (\nabla_y \ell)]$, which gives the approximation $H_I = \mathbb{E}[x^T x]$ and $H_O = \mathbb{E}[(\nabla_y \ell)^T (\nabla_y \ell)]$ (Martens & Grosse, 2015). In Shampoo, the authors propose approximating the empirical Fisher with $H_I = \mathbb{E}[(\nabla_{W^*}\ell)^T(\nabla_{W^*}\ell)], H_O = \mathbb{E}[(\nabla_{W^*}\ell)(\nabla_{W^*}\ell)^T]$ (Gupta et al., 2018). Finally, there exist "eigencorrected" versions of these approximations (George et al., 2018; Vyas et al., 2025), which are beyond this discussion.

## 3. Yet Another Quantization Algorithm

Here, we describe YAQA, a layerwise adaptive rounding method that rounds layers to minimize the full model KL divergence. YAQA consists of two components: 1) a theoretically principled rounding algorithm that generalizes LDLQ to consider the full model error, and 2) a series of near-

optimal Hessian sketches for the rounding algorithm that can be tractably computed for large modern LLMs. Since YAQA only chooses a representation within a quantized space, YAQA can be used with any quantizer. This means that YAQA does not affect the inference efficiency of the quantized model, which is determined by the quantizer.

### 3.1. End-to-End Layerwise Adaptive Rounding

In the state-of-the-art local layerwise adaptive rounding algorithm LDLQ, each linear layer weight matrix $W^* \in \mathbb{R}^{m \times n}$ is independently rounded to produce $W$ with the following fixed point iteration:

$$W = \mathcal{Q}\left(W^* + (W^* - W)(L_1 - I)\right), \qquad (3)$$

where $L_1$ is the triangular component of the LDL decomposition of $H_1$ and specifies linear feedback along the input channels of $W^*$. Here, $\mathcal{Q}$ is a scalar quantizer; we detail the vector quantization case in the Appendix. LDLQ acts to minimize the loss $\text{tr}((W^* - W)H_1(W^* - W)^T)$, which is a proxy for the end-to-end error $\mathcal{L}(W)$ in Equation 2.

We wish to generalize this to other Hessian sketches that give better quadratic proxy losses for $\mathcal{L}(W)$. Consider a Hessian sketch $\tilde{H} \in \mathbb{R}^{mn \times mn}$ with LDL decomposition $\tilde{H} = LDL^T$ and the fixed-point iteration

$$W = \text{vec}^{-1}\left(\mathcal{Q}\left(\text{vec}(W^*) + \text{vec}(W^* - W)(L - I)\right)\right), \qquad (4)$$

where $\text{vec}^{-1}$ returns to the original shape. This update, which requires $\mathcal{O}(m^2 n^2)$ operations as written, is obviously intractable: we are interested in characterizing when the *structure* of $\tilde{H}$ allows adaptive rounding to be efficient.

### 3.1.1. WHEN IS ADAPTIVE ROUNDING FAST?

For Equation 4 to be fast, we need two properties to hold. First, $L$ must admit fast matrix-vector multiplication in $o(m^2 n^2)$ time. Second, the fixed-point iteration should terminate in a small number of steps. Structured matrices that satisfy the former have been well-studied (De Sa et al., 2017), so we focus on the latter. The key property we need is a metric we call the "structural nilpotence degree" (SND):

**Definition 3.1** (SND). Let $L$ be a unit triangular matrix. $\text{snd}(L)$ is the degree of the binary nilpotent matrix $N$ with the same support (nonzero mask) as $L - I$.

The SND bounds the number of steps Equation 4 converges in, which follows since $N = L - I$ is the adjacency matrix of the dependency graph of Equation 4's update.

**Lemma 3.2.** *Equation 3 converges after $\leq \text{snd}(L)$ steps.*

It is straightforward to see that for a general dense lower triangular $L \in \mathbb{R}^{mn \times mn}$, $\text{snd}(L) = mn$, which matches our earlier analysis of running LDLQ on $\tilde{H}$ and $\text{vec}(W^*)$.

Lemma 3.2 reduces the problem of finding a $\tilde{H}$ that makes LDLQ tractable to finding one with fast matrix-vector multiplication and low SND. This next lemma helps us do that.

**Lemma 3.3.** *Let $L_1, L_2$ be unit triangular matrices. Then* $\mathsf{snd}(L_1 \otimes L_2) = \mathsf{snd}(L_1) + \mathsf{snd}(L_2)$.

This result motivates us to use a Kronecker-factored approximation $\tilde{H} = H_O \otimes H_I$, for $H_O \in \mathbb{R}^{m \times m}$ and $H_I \in \mathbb{R}^{n \times n}$, in YAQA. Since the LDL decomposition of this $\tilde{H}$ is $H_O \otimes H_I = (L_O \otimes L_I)(D_O \otimes D_I)(L_O \otimes L_I)^T$, where $L_O D_O L_O^T$ is the LDL decomposition of $H_O$ and likewise for $H_I$, it follows that for $L = L_O \otimes L_I$, $\mathsf{snd}(L) = \mathsf{snd}(L_O) + \mathsf{snd}(L_I) \leq m + n - 1$. We can also see that this admits fast multiplication, as running Equation 4 on this $\tilde{H}$ (hereafter called "YAQA") is equivalent to

$$W = \mathsf{vec}^{-1}\left(\mathcal{Q}(\mathsf{vec}(W^*) + \mathsf{vec}(\Delta)(L_O \otimes L_I - I))\right) \tag{5}$$

$$= \mathcal{Q}(W^* + L_O'^T \Delta L_I' + L_O'^T \Delta + \Delta L_I'), \tag{6}$$

where $L_O' = L_O - I$, $L_I' = L_I - I$, and $\Delta = W^* - W$. This means "YAQA" can run in $m + n$ iterative steps, each of which are small highly parallelizable matmuls.

Compared to LDLQ, Equation 5 has two additional "output side" feedback components: $L_O^T \Delta$ and $L_O^T \Delta L_I$. This is conceptually nice since the feedback is symmetric across input and output channels. Empirically, although quantization time is negligible in practice, YAQA also takes roughly twice as long as LDLQ (SND= $n$) as predicted by their SNDs. In contrast, GuidedQuant, which can also be expressed in our SND framework as LDLQ on a block-diagonal approximation (see Appendix), uses no output side feedback regardless of the number of blocks. Indeed, GuidedQuant's performance plateaus after just 4 blocks.

### 3.1.2. END-TO-END BOUNDS & HESSIAN DESIDERATA

Like LDLQ, we can bound the proxy error of YAQA $\mathsf{vec}(\Delta)\tilde{H}\,\mathsf{vec}(\Delta)^T = \mathsf{tr}(\Delta^T H_O \Delta H_I)$ (see Appendix). However, we can also bound the *"true second-order error"* $\mathsf{vec}(\Delta)H\,\mathsf{vec}(\Delta)^T$, for the "true Hessian" $H = \nabla^2 \mathcal{L}(W^*)$, by the proxy error and cosine similarity between $H_O \otimes H_I$ and $H$, which gives *the first end-to-end error bound for any quantization algorithm*.

**Theorem 3.4.** *Let $H \in \mathbb{R}^{mn \times mn} = \nabla^2 \mathcal{L}(W^*)$ be the Hessian of a linear layer $W$ with respect to the KL to the original model outputs, $H_O \in \mathbb{R}^{m \times m}$ and $H_L \in \mathbb{R}^{n \times n}$ be two p.d. matrices, and $\mathcal{Q}$ perform nearest or stoch. rounding with $\mathbb{E}[(\mathcal{Q}(x) - x)^2] \leq \sigma^2$. Furthermore, let $W$ be the output of Equation 5 with $L_O', L_I'$ from the LDL decomposi-*

*tions of $H_O, H_I$, respectively. Then,*

$$\mathsf{vec}(\Delta)H\,\mathsf{vec}(\Delta)^T \leq \|H\|_F \Big( \|\Delta\|_F^2 \sqrt{2 - 2c} +$$

$$\frac{\mu_I^2 \mu_O^2}{mn \|H_I\|_F \|H_O\|_F} \mathsf{tr}(H_I^{1/2})^2 \, \mathsf{tr}(H_O^{1/2})^2 \sigma^2 \Big)$$

*where $c = \frac{\langle H, H_O \otimes H_I \rangle}{\|H\|_F \|H_O\|_F \|H_I\|_F}$ is the cosine similarity between $H$ and $H_O \otimes H_I$.*

Theorem 3.4 states two things. First, the closer $H_O \otimes H_I$ is directionally to $H$, the better YAQA minimizes Equation 2. This intuitively makes sense since $H$ captures the important rounding directions – having an approximation that is directionally similar to $H$ should give a better quantized model. Second, $H_O$ and $H_I$ should both have low incoherence and, as we will show, be approximately low rank. The former follows directly from $\mu_O$ and $\mu_I$, and the latter follows from the fact that regular LDLQ is equivalent to YAQA with $H_O = I$ and $H_I = H_1$.

With this equivalence, the ratio between the "trace part" of the bounds for YAQA and LDLQ is

$$\frac{\mu_O^2 \mu_I^2 \, \mathsf{tr}(H_I^{1/2})^2 \|H_1\| \mathsf{tr}(H_O^{1/2})^2}{m\sqrt{m} \mu_1^2 \, \mathsf{tr}(H_1^{1/2})^2 \|H_I\| \|H_O\|}. \tag{7}$$

From Cauchy-Schwarz, $\mathsf{tr}(H_O^{1/2})^2 \leq k_O \, \mathsf{tr}(H_O)$, so if $H_O$ has rank $k_O \leq \frac{m\mu_1^2 \, \mathsf{tr}(H_1^{1/2})^2 \|H_I\|}{\mu_O^2 \mu_I^2 \, \mathsf{tr}(H_I^{1/2})^2 \|H_1\|}$ then Equation 7 $\leq 1$. When $H_O$ is low rank, which is empirically true, YAQA achieves a lower error bound than LDLQ.

These properties give us a clear objective for constructing a good Kronecker-factored Hessian sketch for YAQA: we wish to maximize $c$ and minimize the incoherences and ranks of $H_O, H_I$. Although we cannot easily control the rank of either factor, we *can* maximize $c$ with power iteration. Specifically, the Kronecker product is a reshaped rank-1 product (Loan, 2000) so we can find $H_O$ and $H_I$ by power iterating on $H$. In the following section, we describe how to tractably do so for modern LLMs.

### 3.2. Scalable Near-Optimal Kronecker Factored Hessian Sketches

We wish to find $H_I, H_O = \arg\min_{H_I, H_O} \|H - H_O \otimes H_I\|_F^2$. This can be done with power iteration:

$$(H_I)_i \leftarrow H(H_O)_{i-1} / \|(H_O)_{i-1}\|_F^2 \tag{8}$$

$$(H_O)_i \leftarrow (H_I)_{i-1} H / \|(H_I)_{i-1}\|_F^2. \tag{9}$$

which is guaranteed to converge to the optimal $H_O, H_I$. However, power iterating on $H$ is practically difficult. Recall that $H$ is equal to the real Fisher Information Matrix, or $\mathbb{E}[\mathsf{vec}(\nabla_{W^*}\ell)\mathsf{vec}(\nabla_{W^*}\ell)^T]$, where the expectation is taken

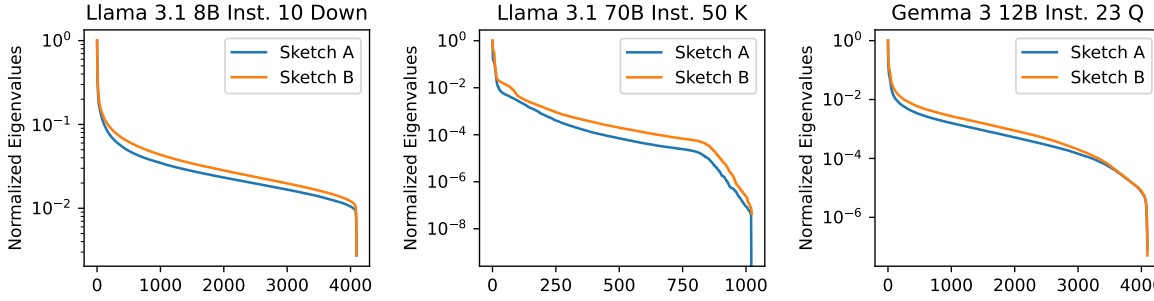

*Figure 1.* Real-world $H_O$'s from A and B are approximately low rank and have similar spectrums.

over independent samples and $\ell$ is the cross entropy with a Monte-Carlo sample from the output logits. In LLMs, tokens within a sequence are not independent due to sequence mixing (e.g. with attention). As such, $H$ must be computed over *entire sequences and not individual tokens*, which increases the variance of the estimate.

To remedy this, we propose two different Hessian sketches that both have high cosine similarity to $H$ while being tractable. Sketch A assumes tokens are independent within a linear layer, which reduces the variance to $O(1/\#\text{tokens})$ at the cost of a slightly biased estimate. Sketch B runs one round of power iteration on $\nabla^2 \mathcal{L}(W^*)$ starting from an identity initialization, which can be computed in a single pass over a dataset. This lets us use enough sequences to achieve low variance without blowing up cost. We show that given sufficient data, Sketch B both theoretically and empirically outperforms Sketch A. However, both are still significantly better than prior state-of-the-art rounding algorithms.

### 3.2.1. HESSIAN SKETCH A

Sketch A power iterates on a Hessian estimate that assumes that tokens within a sequence are independent. In this setting, $(\nabla^2 \mathcal{L}(W^*))_A = \mathbb{E}_{x \sim D} \left[ x^T x \otimes (\nabla_y \ell)^T (\nabla_y \ell) \right]$, where $\ell$ is computed with the same Monte-Carlo sample as before but each $x, y$ pair corresponds to an *individual token*. This sketch is obviously biased but reduces the variance by increasing the sample size to the number of tokens. Furthermore, $(\nabla^2 \mathcal{L}(W^*))_A$ still has sequence information from $(\nabla_y \ell)$, so it is still empirically close to $\nabla^2 \mathcal{L}(W^*)$. To power iterate on $(\nabla^2 \mathcal{L}(W^*))_A$, we compute

$$(H_I)_i \leftarrow \frac{\mathbb{E}_{x \sim D} \left[ x^T x \left\langle (H_O)_{i-1}, (\nabla_y \ell)^T (\nabla_y \ell) \right\rangle \right]}{\|(H_O)_{i-1}\|_F^2} \quad (10)$$

$$(H_O)_i \leftarrow \frac{\mathbb{E}_{x \sim D} \left[ (\nabla_y \ell)^T (\nabla_y \ell) \left\langle (H_I)_{i-1}, x^T x \right\rangle \right]}{\|(H_I)_{i-1}\|_F^2}. \quad (11)$$

Equations 10 and 11 only require the input $x$ and error signal $\frac{d\ell}{dy}$, so we can use a modified backward pass to perform fully-distributed power iteration. To speed up convergence, we initialize $H_O, H_I$ with the LDLQ Hessian: $(H_I)_0 \leftarrow H_1, (H_O)_0 \leftarrow I$. Empirically, Sketch A converges in $\leq 3$ full iterations and takes around 20 GPU-hours

for a 10B parameter model and 20M token dataset, although it is possible to significantly fewer sequences without affecting quality.

### 3.2.2. HESSIAN SKETCH B

Sketch B directly computes the result of one round of power iteration on $\nabla^2 \mathcal{L}(W^*)$ starting with $H_I, H_O \leftarrow I$. Power iterating on $\nabla^2 \mathcal{L}(W^*)$ computes the following updates

$$(H_I)_i \leftarrow \frac{\mathbb{E}_{s \sim D} \left[ (\nabla_{W^*} \ell)^T (H_O)_{i-1} (\nabla_{W^*} \ell) \right]}{\|(H_O)_{i-1}\|_F^2} \quad (12)$$

$$(H_O)_i \leftarrow \frac{\mathbb{E}_{s \sim D} \left[ (\nabla_{W^*} \ell)(H_I)_{i-1} (\nabla_{W^*} \ell)^T \right]}{\|(H_I)_{i-1}\|_F^2}. \quad (13)$$

If $(H_I)_0$ and $(H_O)_0$ are both $I$, then $(H_I)_1 = \mathbb{E}_{s \sim D} \left[ (\nabla_{W^*} \ell)^T (\nabla_{W^*} \ell) \right] / m$ and $(H_O)_1 = \mathbb{E}_{s \sim D} \left[ (\nabla_{W^*} \ell)(\nabla_{W^*} \ell)^T \right] / n$, where the expectation and $\nabla_{W^*} \ell$ are computed over sequences. Both of these are computable in the same backward pass, letting us do one round of power iteration on both $H_O$ and $H_I$ in a *single pass over a dataset*. This sketch is conceptually similar to the preconditioning basis in Shampoo (Gupta et al., 2018; Morwani et al., 2025), except that we compute the gradient per-sequence instead of per-batch and use the real Fisher instead of the empirical Fisher. Sketch B takes around 30 GPU-hours for a 10B model and 64K 2K-token sequences. Again, like with Sketch A, it is possible to use far fewer sequences (as few as 2K, or 1 GPU-hour) while still achieving state of the art results (see Appendix).

### 3.2.3. EVALUATING SKETCH A AND B

In Section 3.1, we desired to find a Kronecker-factored sketch with low rank, low incoherence, and high cosine similarity to the true Hessian. Here, we evaluate how well Sketch A and B achieve these goals. Figure 1 shows that empirically, real-world $H_O$ matrices have strong spectral decay and are approximately low rank. Figure 2 shows the empirical CDF across layers of the ratio $\frac{\text{tr}(D_O^{\text{IP}}) \text{tr}(D_I^{\text{IP}})}{\text{tr}(D_O) \text{tr}(D_I)}$, where $D_I^{\text{IP}}$ denotes $D_I$ after incoherence processing $H_I$ with the RHT and likewise for $D_O^{\text{IP}}$ and $H_O$. By applying IP, we can bound the incoherences of $H_O$ and $H_I$, which

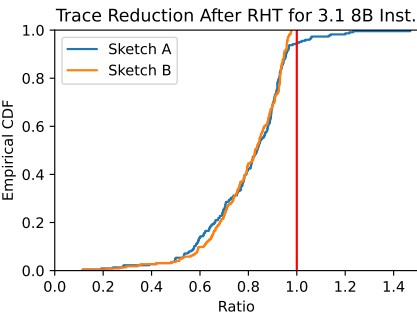

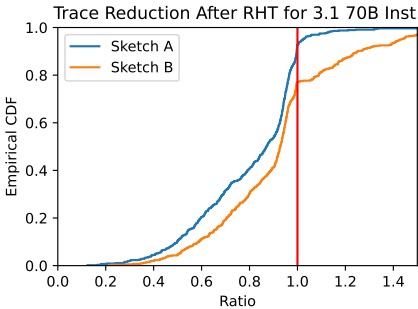

*Figure 2.* Empirical CDF of $\frac{\mathrm{tr}(D_O^{\mathrm{IP}})\,\mathrm{tr}(D_I^{\mathrm{IP}})}{\mathrm{tr}(D_O)\,\mathrm{tr}(D_I)}$ across linear layers, where $D^{\mathrm{IP}}$ denotes $D$ after incoherence processing $H$ with the RHT. A ratio of $< 1$ indicates a reduction in error bound.

empirically translates to a lower trace bound.

Figure 3 L and C contain plots of the "normalized" cosine similarity of A, B, and $I \otimes H_1$ (LDLQ's Hessian). Here, the normalized cosine similarity denotes $\frac{\langle H, H_O \otimes H_I \rangle}{\|H_O\|\|H_I\|}$, scaled so the largest data point is 1.0. This avoids computing $\|H\|$, which is unnecessary. The cosine similarities of A and B are both much higher than $I \otimes H_1$, with A being slightly lower than B. In certain data-limited cases, such as for layer 24 Down in Llama 3.1 8B Instruct, A has a higher cosine similarity than B. Here, the bias of A is less than the variance of B. Finally, Figure 3 R shows that A and B have high cosine similarities to each other, with the pre-attention score layers (Q, K) having higher similarity than the post-attention score layers. This correlates with our construction of A, which ignores sequence mixing. Indeed, across all desiderata, YAQA is provably better than LDLQ.

## 4. Experiments

To test YAQA, we quantized Llama, Gemma, and Qwen models and evaluated both downstream performance (perplexity, zeroshot accuracy) and the KL divergence to the original model outputs. Llama and Gemma results can be found here in the main body and Qwen results are in the Appendix. The KL considers the entire distribution over the output space, whereas perplexity and zeroshot accuracy only consider the mass of a single "ground truth" token (see Appendix). Since our goal is to produce a model as close to the original model as possible and almost all real-world use cases of LLMs involve sampling from the entire distribution, the KL is better at measuring how close models are.

**Baselines and Setup** The focus of YAQA is on the Hessian estimate and weight rounding algorithm, so our main experiments perform *weight-only* PTQ without any additional finetuning. However, we include experiments with recovery finetuning to show that YAQA composes with finetuning. For YAQA-A, we use a context length of 8K, 20M tokens, and 3 full rounds of power iteration. For YAQA-B, we use a context length of 2K and 64K sequences.

In the following sections, we compare against LDLQ (Chee et al., 2023), GuidedQuant (Kim et al., 2025), DiscQuant (Chee et al., 2025), and Google's Gemma 3 QAT recipe (Team et al., 2025), which represent state-of-the-art algorithms that perform local adaptive rounding, end-to-end adaptive rounding, constrained gradient-based rounding, and QAT, respectively. LDLQ is equivalent to GPTQ and *is the rounding algorithm used in almost all SOTA weight-only quantization works*, including GPTQ, QuIP, QuIP#, QTIP, as well as weight-activation quantization works such as QuaRot, SpinQuant, and NestQuant (Chee et al., 2023; Tseng et al., 2024a;b; Ashkboos et al., 2024; Liu et al., 2025; Savkin et al., 2025). Of our baselines, only LDLQ does not require computing gradients, and QAT requires significantly more data than the others. Regardless, YAQA achieves a lower KL than *all presented baselines* while achieving state of the art downstream performance.

We do not compare to works such as AQLM, AWQ, and SmoothQuant, since the above LDLQ-based methods already outperform them (Egiazarian et al., 2024; Lin et al., 2023; Xiao et al., 2024). Likewise, we do not compare to older QAT methods like LSQ and LLM-QAT, since these do not consistently outperform *existing* LDLQ-based baselines and are conceptually different than the "continued pretraining" QAT that modern industry labs use (Esser et al., 2020; Liu et al., 2023; Bondarenko et al., 2024; Team et al., 2025). Additional experiments and comparisons to CBQ (Ding et al., 2025) and PV-Tuning (Malinovskii et al., 2024) are in the Appendix.

### 4.1. Second Order Quantization Algorithms

Table 1 compares YAQA against LDLQ and GuidedQuant, which respectively use local and global second order information during adaptive rounding. YAQA consistently outperforms both LDLQ and GuidedQuant in all metrics, with YAQA-B being better than YAQA-A as expected. In Section 3.2.3, we showed that LDLQ was provably worse than YAQA; indeed, this is reflected empirically. GuidedQuant uses a block diagonal approximation of the empirical Fisher and runs LDLQ on each block. Although GuidedQuant outperforms LDLQ and prior "end-to-end" methods like

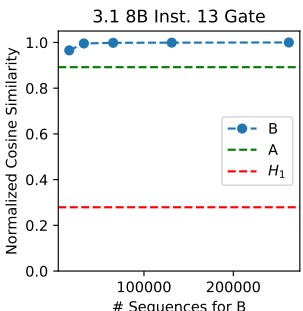 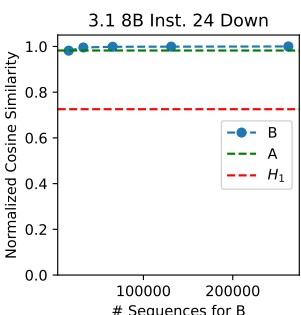 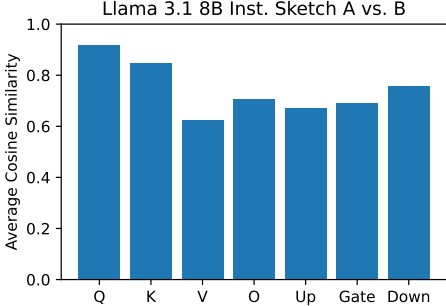

*Figure 3.* (L, C) Normalized cosine similarity of $I \otimes H_1$, A, and different sequence counts for B, calculated against $H$. A and B are both much closer to $H$ than $I \otimes H_1$. (R) Average cosine similarity between A and B, grouped by linear layer.

*Table 1.* Results with incoherence processing, the QTIP quantizer, and no finetuning. YAQA strongly outperforms LDLQ and GuidedQuant, two SOTA local and global adaptive rounding methods. YAQA reduces the KL by 30% over LDLQ. Individual 0-shot results in Appendix.

| ALGO. | BITS | $D_{KL} \downarrow$ | PPL $\downarrow$ | | 0-SHOT $\uparrow$ | $D_{KL} \downarrow$ | PPL $\downarrow$ | | 0-SHOT $\uparrow$ |
|---|---|---|---|---|---|---|---|---|---|
| | | W2 | W2 | C4 | AVG | W2 | W2 | C4 | AVG |
| | | 3.1 70B INST. | | | | 3.2 3B INST. | | | |
| BF16 | 16 | 0 | 3.52 | 6.46 | 67.67 | 0 | 9.58 | 10.62 | 63.79 |
| LDLQ | 2 | 0.497 | 6.02 | 7.82 | 65.45 | 0.455 | 15.30 | 14.69 | 55.68 |
| | 3 | 0.138 | 4.26 | 6.74 | 67.58 | 0.085 | 10.69 | 11.44 | 62.46 |
| | 4 | 0.045 | 3.74 | 6.54 | 67.58 | 0.021 | **9.78** | 10.79 | 63.70 |
| GUIDEDQUANT | 2 | 0.383 | 5.59 | 7.51 | 65.44 | 0.342 | 13.56 | 13.96 | 57.72 |
| | 3 | 0.111 | 4.10 | 6.68 | 67.33 | 0.070 | 10.48 | 11.19 | 63.48 |
| | 4 | 0.035 | 3.72 | 6.53 | **67.80** | 0.019 | 9.84 | 10.80 | **64.10** |
| **YAQA-A** | 2 | 0.378 | 5.56 | 7.51 | 65.92 | 0.333 | 13.75 | 13.56 | 58.29 |
| | 3 | 0.110 | 4.10 | 6.68 | **67.69** | 0.059 | 10.37 | 11.14 | 62.51 |
| | 4 | 0.036 | 3.73 | 6.52 | 67.50 | 0.015 | **9.78** | 10.76 | 63.28 |
| YAQA-B | 2 | **0.335** | **5.30** | **7.34** | **66.19** | **0.288** | **13.18** | **13.10** | **59.11** |
| | 3 | **0.094** | **4.01** | **6.64** | 67.24 | **0.047** | **10.15** | **11.09** | 62.74 |
| | 4 | **0.030** | **3.69** | **6.51** | 67.73 | **0.014** | 9.80 | **10.75** | 63.31 |
| | | 3.1 8B INST. | | | | 3.2 1B INST. | | | |
| BF16 | 16 | 0 | 6.50 | 8.02 | 69.82 | 0 | 11.57 | 13.20 | 54.79 |
| LDLQ | 2 | 0.356 | 9.39 | 10.70 | 62.51 | 0.527 | 19.86 | 19.66 | 49.60 |
| | 3 | 0.069 | 7.05 | 8.50 | 69.29 | 0.109 | 12.95 | 14.44 | 52.89 |
| | 4 | 0.019 | 6.63 | 8.15 | 69.41 | 0.032 | 12.05 | 13.63 | 53.47 |
| GUIDEDQUANT | 2 | 0.317 | 9.03 | 10.42 | 63.12 | 0.473 | 18.95 | 19.54 | 48.76 |
| | 3 | 0.062 | 6.98 | 8.44 | 68.53 | 0.095 | 12.82 | 14.33 | 53.48 |
| | 4 | 0.018 | 6.65 | 8.14 | 69.05 | 0.027 | 11.93 | 13.54 | **54.18** |
| **YAQA-A** | 2 | 0.284 | 8.79 | 10.09 | 64.06 | 0.371 | 17.22 | 17.64 | 50.02 |
| | 3 | 0.050 | 6.89 | 8.38 | 69.09 | 0.072 | 12.56 | 14.02 | **53.86** |
| | 4 | 0.015 | 6.62 | 8.13 | 69.62 | 0.021 | **11.83** | 13.44 | 54.17 |
| YAQA-B | 2 | **0.241** | **8.39** | **9.83** | **64.32** | **0.334** | **16.66** | **17.36** | **50.90** |
| | 3 | **0.044** | **6.85** | **8.34** | **69.31** | **0.065** | **12.51** | **13.97** | 53.41 |
| | 4 | **0.013** | **6.61** | **8.12** | **69.78** | **0.019** | **11.83** | **13.41** | 53.84 |

SqueezeLLM, neither its Hessian or rounding algorithm come with easy-to-compute bounds on the end-to-end error. In contrast, YAQA both comes with theoretical guarantees and empirically outperforms GuidedQuant.

## 4.2. First Order Quantization Methods

Tables 2 and 3 compare YAQA with descent-style approaches. Since these methods require requantization or projection at every step, they are effectively limited to sub-

optimal scalar quantizers. In contrast, YAQA is a general rounding algorithm that can be used with any quantizer. Regardless, even when used with scalar quantizers, YAQA outperforms these methods. Table 2 compares YAQA to DiscQuant, which performs gradient descent on the KL in a localized subspace. Both YAQA and LDLQ outperform DiscQuant, showing that unlike YAQA, descent-based methods do not always outperform local adaptive rounding. Table 3 compares YAQA against Google's QAT Gemma 3 12B Instruct (Team et al., 2025). We estimate the QAT process

*Table 2.* Results with incoherence processing, the INT4 quantizer, and no finetuning for Llama 3.1 8B Instruct. YAQA is quantizer agnostic and works with standard datatypes such as INT4.

| ALGO. | $D_{KL} \downarrow$ | PPL $\downarrow$ | | 0-SHOT ACC $\uparrow$ | | | | | |
|---|---|---|---|---|---|---|---|---|---|
| | W2 | W2 | C4 | AVG | ARCC | ARCE | BOOLQ | HSWAG | PIQA |
| BF16 | 0 | 6.50 | 8.02 | 69.82 | 51.37 | 78.03 | 82.05 | 57.74 | 79.92 |
| LDLQ | 0.038 | 6.76 | 8.26 | 67.99 | 50.00 | 76.94 | 77.01 | 56.83 | 79.16 |
| DISCQUANT | 0.061 | 6.83 | 8.37 | 67.68 | 50.34 | 77.44 | 74.12 | 56.55 | 79.92 |
| YAQA-A | **0.028** | **6.71** | **8.21** | **69.11** | 50.68 | 78.11 | 79.65 | 57.13 | 79.98 |
| YAQA-B | 0.029 | 6.72 | 8.23 | 68.92 | 49.49 | 77.31 | 81.01 | 56.98 | 79.82 |

*Table 3.* Results for Gemma 3 12B Inst. with INT4 *without finetuning*. Despite being trained on the original model's outputs, the QAT model has a higher KL to the original model than YAQA.

| ALGO. | QUANT. TYPE | BITS | $D_{KL} \downarrow$ | PPL $\downarrow$ | | 0-SHOT ACC $\uparrow$ | | | | | |
|---|---|---|---|---|---|---|---|---|---|---|---|
| | | | W2 | W2 | C4 | AVG | ARCC | ARCE | BOOLQ | HSWAG | PIQA |
| BF16 | NONE | 16 | 0 | 7.85 | 8.61 | 70.22 | 54.01 | 78.79 | 87.25 | 54.27 | 76.77 |
| GOOGLE QAT | QAT | 4.5 | 0.089 | 7.56 | 8.52 | 70.83 | 54.52 | 79.76 | 86.82 | 54.77 | 78.29 |
| **YAQA-A** | PTQ | 4 | 0.058 | 7.96 | 8.69 | 70.12 | 53.90 | 78.83 | 87.09 | 53.68 | 77.09 |
| **YAQA-B** | PTQ | 4 | 0.056 | 7.94 | 8.67 | 69.90 | 54.10 | 78.66 | 86.91 | 54.13 | 75.68 |

*Table 4.* Results with the QTIP quantizer, incoherence processing, and recovery finetuning. Although finetuning reduces the gap between YAQA and LDLQ, YAQA still achieves state-of-the-art results.

| ALGO. | BITS | $D_{KL} \downarrow$ | PPL $\downarrow$ | | 0-SHOT $\uparrow$ | $D_{KL} \downarrow$ | PPL $\downarrow$ | | 0-SHOT $\uparrow$ |
|---|---|---|---|---|---|---|---|---|---|
| | | W2 | W2 | C4 | AVG | W2 | W2 | C4 | AVG |
| | | LLAMA 3.1 70B INSTRUCT | | | | LLAMA 3.1 8B INSTRUCT | | | |
| BF16 | 16 | 0 | 3.52 | 6.46 | 67.67 | 0 | 6.50 | 8.02 | 69.82 |
| LDLQ | 2 | 0.302 | 5.01 | 7.16 | 66.11 | 0.185 | 7.82 | 9.20 | 65.44 |
| | 3 | 0.101 | 3.96 | 6.64 | **67.46** | 0.048 | 6.80 | 8.31 | 68.42 |
| | 4 | 0.036 | 3.71 | 6.54 | 67.64 | 0.016 | 6.61 | 8.13 | 69.47 |
| **YAQA-A** | 2 | 0.279 | 4.92 | 7.10 | 66.26 | 0.163 | 7.63 | 9.06 | **67.54** |
| | 3 | 0.098 | 3.88 | 6.63 | 66.94 | 0.042 | 6.78 | 8.28 | **69.23** |
| | 4 | 0.032 | 3.68 | **6.52** | 67.59 | 0.014 | 6.58 | **8.10** | 69.50 |
| **YAQA-B** | 2 | **0.266** | **4.82** | **7.07** | **66.99** | **0.147** | **7.60** | **8.96** | 66.38 |
| | 3 | **0.091** | **3.87** | **6.61** | 67.42 | **0.038** | **6.74** | **8.27** | 68.88 |
| | 4 | **0.029** | **3.67** | **6.52** | **67.69** | **0.012** | **6.56** | 8.11 | **70.12** |

took $100\times$ more data than YAQA. Although the QAT model somehow outperforms the original model in downstream tasks, *even without finetuning*, YAQA models have a lower KL divergence to the original model. YAQA also maintains a smaller gap in downstream performance, suggesting that QAT actually produces a considerably different model and YAQA better preserves the original model.

### 4.3. Finetuning

Recent PTQ works have included a "recovery finetuning" step that adjusts unquantized parameters *before quantization* to compensate for the effect of *already-quantized layers* (Tseng et al., 2024a;b; Egiazarian et al., 2024; Malinovskii et al., 2024). This effectively includes cross layer information, which raises the question: how much overlap is there between finetuning and YAQA? Table 4 shows an ex-

periment with QuIP#'s recovery finetuning algorithm and the setup in Table 1. For both models, recovery finetuning reduces the gap between LDLQ and YAQA. However, YAQA still reduces the KL by $\approx$ 10-20% over LDLQ and maintains a large gap in downstream tasks, indicating that YAQA uses global information that is not available during recovery finetuning. Finally, the LDLQ results in this table correspond to the results in the state-of-the-art QTIP paper, so YAQA's results Table 4 set a new state of the art across all PTQ methods and quantizers.

## 5. Conclusion

In this work, we present YAQA, a new rounding algorithm that generalizes state-of-the-art local layerwise quantization methods to minimize the end-to-end error. YAQA introduces a series of theoretical results that give, for the first

time, end-to-end error bounds for quantization algorithms. These results admit a natural Kronecker-factored Hessian structure for use with adaptive rounding algorithms, as well as corresponding near-optimal Hessian sketches. YAQA is provably better than GPTQ/LDLQ and empirically reduces the KL by $\approx 30\%$ over these methods. Even further, YAQA achieves a lower KL than QAT and sets a new empirical state-of-the-art in downstream tasks among PTQ methods, all while adding no inference overhead.

## Impact Statement

This paper presents work whose goal is to advance the field of Machine Learning. There are many potential societal consequences of our work, none which we feel must be specifically highlighted here.

## Acknowledgements

AT was supported by the NSF Graduate Research Fellowship. CD was supported by DARPA YFA D24AP00259 and NSF Career Award 2046760. We thank Together AI for compute resources.

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

# A. Appendix

## A.1. Experimental Setup and Implementation Details

All Hessians were collected using the RedPajama v1 dataset (Computer, 2023). We use Hadamard matrices from Neil Sloane's website for the "base Hadamard matrix" in incoherence processing as described in Section A.5 (Sloane). We use the OPTQ "Wikitext2" and "C4" dataset splits for KL divergence and perplexity calculation (Frantar et al., 2023). We use a sequence length of 8192 for all KL divergence and perplexity evaluations. We evaluate all zeroshot results with the chat template applied. For finetuning experiments, we use the same setup as QTIP except that we normalize the activation error due to numerical instability from the default Adam $\epsilon = 10^{-8}$. For all Llama 3.1 70B Instruct experiments, we do not quantize the `o_v` layer. The Gemma QAT comparison was run with the `google/gemma-3-12b-it-qat-q4_0-gguf` model on Huggingface, which was dequantized with the `4.52.0.dev0` nightly version of Transformers. Our code can be found at `https://github.com/Cornell-RelaxML/yaqa-quantization`.

## A.2. Hessian Sketch Memory and Compute Requirements

YAQA requires storing $O(n^2)$ memory for $H_I$ and $O(m^2)$ memory for $H_O$. In practice, we only need to store the lower triangular parts of $H_I$ and $H_O$ since they are symmetric. LDLQ and other layerwise activation methods only need to store $H_I$, so YAQA requires roughly double the storage for the Hessians. We found that computing and storing the Hessians in FP32 was sufficient to maintain numerical stability and positive-definiteness with a small regularization factor on the diagonal ($\approx 10^{-4} \operatorname{tr}(H)/n$), but using TF32 for computations was not. The computational cost of computing Hessians with Sketch B is $O(btmn + bn^2m + bm^2n)$ on top of the cost of a forward pass and backprop, where $b$ is the number of sequences and $t$ is the average number of tokens per sequence. The computational cost of computing Hessians with Sketch A is $O(pbt(n^2 + m^2))$, again on top of a forward pass and backward pass, where $p$ is the number of steps of power iteration. Since we must use a larger $b$ for Sketch B than Sketch A to get acceptable variance, Sketch B is empirically slower than Sketch A.

## A.3. GuidedQuant as LDLQ

In GuidedQuant, each of $g$ groups of output channels in $W^*$ shares an input Hessian for LDLQ. This corresponds to a block-diagonal Hessian approximation, where there are $g \leq m$ unique $n \times n$ blocks. Since the L part of the LDL decomposition of a matrix is block lower unit triangular, observe that the SND of this approximation is simply the largest SND of any given block. This is $n$, which matches an optimal implementation of GuidedQuant where LDLQ is run in parallel across groups. However, observe that even though the number of serial steps is $n$, the number of FLOPs is increased by a factor of $g$, which means that GuidedQuant is still more expensive than regular LDLQ when $g > 1$.

## A.4. KL Divergence vs. Perplexity

While the full model KL divergence and perplexity are conceptually related, they measure two fundamentally different quantities. The KL measures the difference between two distributions and is defined over the full support of the distributions as $D_{\mathrm{KL}}(p\|q) = \sum_{x \in \mathcal{X}} p(x) \log \frac{p(x)}{q(x)}$. The perplexity measures the mass on a "ground truth" target $\tau$ in single probability distribution $p$: $1/p(\tau)$. As such, two models can have very similar perplexities but be completely different from each other. For example, Llama 1 7B has a Wikitext 2 perplexity of 5.68 and Llama 2 7B has a perplexity of 5.47, but were pretrained from scratch separately. Indeed, their KL divergence is 0.197, which is much higher than what the difference in perplexity would suggest ($\log\left(\frac{5.68}{5.47}\right) = 0.038$)

## A.5. Incoherence Processing with the Random Hadamard Transform

Although we describe incoherence processing for Hessians in the main body, incoherence processing also modifies the weights. The full definition of incoherence, including *weight incoherence*, is as follows:

**Definition A.1** (Chee et al. (2023)). A Hessian $H \in \mathbb{R}^{n \times n}$ is $\mu$-incoherent if its eigendecomposition $H = Q\Lambda Q^T$ has $\max_{i,j} |Q_{ij}| = \max_{i,j} |e_i^T Q e_j| \leq \mu/\sqrt{n}$. A weight matrix $W \in \mathbb{R}^{m \times n}$ is $\mu$-incoherent if $\max_{i,j} |W_{ij}| = \max_{i,j} |e_i^T W e_j| \leq \mu \|W\|_F/\sqrt{mn}$.

Incoherence processing with the RHT applies a Random Hadamard Transformation on $W$ and $H$. Let H be a Hadamard matrix, defined as follows:

$$H_1 = \begin{bmatrix} 1 \end{bmatrix} \qquad\qquad H_n = \frac{1}{\sqrt{2}} \begin{bmatrix} H_{n-1} & H_{n-1} \\ H_{n-1} & -H_{n-1} \end{bmatrix} \qquad\qquad (14)$$

Then, the RHT performs $x \leftarrow H_{\log_2 n} S x$, where $S$ is a random sign vector $\in \{\pm 1\}^n$, $x \in \mathbb{R}^n$, and $n$ is a power of 2. For non power-of-2 $n$, we follow QuIP# and use a fixed "small" Hadamard matrix as a base instead of $H_1$. Full proofs for bounds on the behavior of the RHT can be round in QuIP#.

### A.6. YAQA Rounding Algorithm

The YAQA rounding algorithm can be written as a fixed point iteration (Algorithm 1). It can also be implemented iteratively (Python code), which is faster for slower quantizers than the fixed point iteration implementation above.

---

**Algorithm 1** YAQA Rounding Algorithm Fixed Point Iteration

---

**Require:** Weight matrix $W \in \mathbb{R}^{m \times n}$, p.d. $H_O \in \mathbb{R}^{m \times m}$, p.d. $H_I \in \mathbb{R}^{n \times n}$, quantizer $\mathcal{Q}$, quantizer block sizes $g_x \in \mathbb{Z}^+|m, g_y \in \mathbb{Z}^+|n$.

1:  $L_O, D_O \leftarrow \text{BlockLDL}(H_O, g_x)$
2:  $L_I, D_I \leftarrow \text{BlockLDL}(H_I, g_y)$
3:  $\hat{W} \leftarrow \mathcal{Q}(W)$
4:  converged $\leftarrow$ FALSE
5:  $\hat{W}^* \leftarrow \hat{W}$
6:  **while** $\neg$ converged **do**
7:      $\Delta W \leftarrow W - \hat{W}$
8:      $\hat{W} = \mathcal{Q}(W + L_O^T \Delta W L_I + L_O^T \Delta W + \Delta W L_I)$
9:      converged $\leftarrow (\hat{W} == \hat{W}^*)$
10:     $\hat{W}^* \leftarrow \hat{W}$
11: **end while**
**output** Quantized $\hat{W}$.

---

```python
def YAQA_iterative(W, Lin, Lout, td_x, td_y, cb):
    m, n = W.shape
    hatW = torch.zeros_like(W)
    Qidxs = torch.zeros(m, n, dtype=cb.idx_dtype, device=W.device)
    assert m % td_x == 0 and n % td_y == 0
    starts = [*[(m//td_x-i-1, n//td_y-1) for i in range(m//td_x)], *[(0, n//td_y-i-1) for i in range(n//td_y)]]

    for i in tqdm(range(m//td_x + n//td_y)):
        target = []
        target_idx = []
        start = starts[i]
        jmax = max(start[0], start[1])
        jm, jn = start
        while 0 <= jm < m//td_x and 0 <= jn < n//td_y:
            thing = W[jm*td_x:(jm+1)*td_x, jn*td_y:(jn+1)*td_y] + (Lout[jm*td_x:, jm*td_x:(jm+1)*td_x].T @ (W[jm*
td_x:, jn*td_y:] - hatW[jm*td_x:, jn*td_y:]) @ Lin[jn*td_y:, jn*td_y:(jn+1)*td_y] + Lout[jm*td_x:, jm*td_x:(jm
+1)*td_x].T @ (W[jm*td_x:, jn*td_y:(jn+1)*td_y] - hatW[jm*td_x:, jn*td_y:(jn+1)*td_y]) + (W[jm*td_x:(jm+1)*td_x
, jn*td_y:] - hatW[jm*td_x:(jm+1)*td_x, jn*td_y:]) @ Lin[jn*td_y:, jn*td_y:(jn+1)*td_y])
            target.append(thing)
            target_idx.append((jm, jn))
            jm += 1
            jn -= 1

        target = torch.stack(target, dim=0).reshape(-1, td_x * td_y)
        qtarget, target_idx = cb.quantize(target)
        qtarget = qtarget.reshape(-1, td_x, td_y)
        target_idx = target_idx.reshape(-1, td_x, td_y)

        for j in range(len(target_idx)):
            jm, jn = target_idx[j]
            hatW[jm*td_x:(jm+1)*td_x, jn*td_y:(jn+1)*td_y] = qtarget[j]
            Qidxs[jm*td_x:(jm+1)*td_x, jn*td_y:(jn+1)*td_y] = target_idx[j]

    return hatW, Qidxs
```

## A.7. Proofs

**Lemma 3.2.** *Equation 3 converges after $\leq \mathsf{snd}(L)$ steps.*

*Proof.* Consider the binary matrix $B$ with the same support as $L - I$. This forms an adjacency matrix for the dependency graph of LDLQ, where $B_{i,j} = 1$ iff weight $i$ is rounded with feedback from weight $j$. Since $B$ has degree $k = \mathsf{snd}(L)$, the DAG corresponding to the dependency graph has depth equal to $k$. Each step of LDLQ corresponds to advancing one level in the dependency DAG, so LDLQ converges in $k$ steps. □

**Lemma 3.3.** *Let $L_1, L_2$ be unit triangular matrices. Then $\mathsf{snd}(L_1 \otimes L_2) = \mathsf{snd}(L_1) + \mathsf{snd}(L_2)$.*

*Proof.* Suppose that $N_1^{k_1} = (L_1 - I)^{k_1} = 0$ and similarly for $N_2$ and $k_2$. Then, if $k = k_1 + k_2 - 1$,

$$(L_1 \otimes L_2 - I)^k = ((L_1 - I) \otimes (L_2 - I) + (L_1 - I) \otimes I + I \otimes (L_2 - I))^k \tag{15}$$

$$= (N_1 \otimes N_2 + N_1 \otimes I + I \otimes N_2)^k \tag{16}$$

$$= \sum_{i,j} \binom{k}{i,j} (N_1 \otimes N_2)^{k-i-j} \cdot (N_1 \otimes I)^i \cdot (I \otimes N_2)^j \tag{17}$$

$$= \sum_{i,j} \binom{k}{i,j} \left(N_1^{k-j} \otimes N_2^{k-i}\right), \tag{18}$$

where we can apply the multinomial theorem here to this matrix power because all these matrices ($N_1 \otimes N_2$, $N_1 \otimes I$, and $I \otimes N_2$) commute. But since this sum goes over $i + j \leq k$, it must be that $(k - j) + (k - i) = 2k - i - j \geq k = k_1 + k_2 - 1$. But this means that either $k - j \geq k_1$, or $k - i \geq k_2$ (since otherwise $k - j \leq k_1 - 1$ and $k - i \leq k_2 - 1$ and summing yields a contradiction). So then either $N_1^{k-j}$ or $N_2^{k-i}$ must be zero in each of these terms, and so we get $(L_1 \otimes L_2 - I)^k = 0$. □

**Theorem A.2.** *Let $H_O \in \mathbb{R}^{m \times m}$ and $H_L \in \mathbb{R}^{n \times n}$ be two positive definite matrices and let $\mathcal{Q}$ perform nearest or stochastic rounding independently on blocks of $g_x \times g_y$ with $\mathbb{E}[(\mathcal{Q}(vec(x) - vec(x)))(\mathcal{Q}(vec(x)) - vec(x))^T] \preceq \sigma^2 I$. Furthermore, let $W$ be the output of Equation 5 with $L_O$ and $L_I$ from the $g_x$ and $g_y$-block LDL decompositions of $H_O$ and $H_I$, respectively. Then,*

$$\Delta W (H_O \otimes H_I) \Delta W^T \leq \mathrm{tr}(D_I) \, \mathrm{tr}(D_O) g_x g_y \sigma^2 \leq \frac{g_x g_y \mu_I^2 \mu_O^2}{mn} \mathrm{tr}(H_I^{1/2})^2 \, \mathrm{tr}(H_O^{1/2})^2 \sigma^2$$

*where $\Delta W = W^* - W$ and $\mu_O, \mu_I$ are the incoherences of $H_O, H_I$ (Definition A.1).*

*Proof.* Let

$$\eta = W^* + L_O^T \Delta W L_I + L_O^T \Delta W + \Delta W L_I - W \tag{19}$$

$$= (L_O + I)^T \Delta W (L_I + I) \tag{20}$$

Then,

$$\Delta W (H_O \otimes H_I) \Delta W^T = \mathrm{tr}(\Delta W H_I \Delta W^T H_O) \tag{21}$$

$$= \mathrm{tr}(\Delta W (L_I + I) D_I (L_I + I)^T \Delta W^T (L_O + I) H_O (L_O + I)^T) \tag{22}$$

$$= \mathrm{tr}(\eta D_I \eta^T D_O) = \mathrm{tr}(\eta^T \eta (D_O \otimes D_I)) \tag{23}$$

Since

$$\eta = W^* + L_O^T \Delta W L_I + L_O^T \Delta W + \Delta W L_I - W \tag{24}$$

$$= W^* + L_O^T \Delta W L_I + L_O^T \Delta W + \Delta W L_I - \mathcal{Q}(W^* + L_O^T \Delta W L_I + L_O^T \Delta W + \Delta W L_I) \tag{25}$$

$$= \star - Q(\star) \tag{26}$$

and $\mathcal{Q}$ operates independently on $g_x \times g_y$-sized blocks, we have that $\operatorname{tr}(\eta^T \eta (D_O \otimes D_I)) \leq g_x g_y \sigma^2 \operatorname{tr}(D_O \otimes D_I) = g_x g_y \sigma^2 \operatorname{tr}(D_O) \operatorname{tr}(D_I)$. From Chee et al. (2023), we have that

$$\operatorname{tr}(D) \leq \frac{\mu}{k} \operatorname{tr}(H^{1/2})^2 \tag{27}$$

for arbitrary p.d. $H \in \mathbb{R}^{k \times k}$, so

$$\operatorname{tr}(D_O) \operatorname{tr}(D_I) g_x g_y \sigma^2 \leq \frac{g_x g_y \mu_I^2 \mu_O^2}{mn} \operatorname{tr}(H_I^{1/2})^2 \operatorname{tr}(H_O^{1/2})^2 \sigma^2 \tag{28}$$

$\square$

**Theorem A.3.** *Let $H_O \in \mathbb{R}^{m \times m}$ and $H_L \in \mathbb{R}^{n \times n}$ be two positive definite matrices and let $\mathcal{Q}$ perform nearest or stochastic rounding with $\mathbb{E}[(\mathcal{Q}(x) - x)^2] \leq \sigma^2$. Furthermore, let $W$ be the output of Equation 5 with $L_O$ and $L_I$ from the LDL decompositions of $H_O$ and $H_I$, respectively. Then,*

$$\Delta W (H_O \otimes H_I) \Delta W^T \leq \operatorname{tr}(D_I) \operatorname{tr}(D_O) \sigma^2 \leq \frac{\mu_I^2 \mu_O^2}{mn} \operatorname{tr}(H_I^{1/2})^2 \operatorname{tr}(H_O^{1/2})^2 \sigma^2$$

*where $\Delta W = W^* - W$ and $\mu_O, \mu_I$ are the incoherences of $H_O, H_I$ (Definition A.1).*

*Proof.* This follows from setting $g_x = g_y = 1$ in Theorem A.2. $\square$

**Lemma A.4.** *Let $A, B \in \mathbb{R}^{n \times n}$ be positive definite matrices and $x \in \mathbb{R}^n$ be a vector. Then, $\left| x^T \frac{A}{\|A\|} x - x^T \frac{B}{\|B\|} x \right| \leq \|x\|_F^2 \sqrt{2 - 2c}$ where $\frac{\langle A, B \rangle}{\|A\|\|B\|} = c$.*

*Proof.*

$$\left| x^T \frac{A}{\|A\|} x - x^T \frac{B}{\|B\|} x \right| = \left| x^T \left( \frac{A}{\|A\|} - \frac{B}{\|B\|} \right) x \right| \tag{29}$$

$$\leq \left\| \frac{A}{\|A\|} - \frac{B}{\|B\|} \right\|_F \|x\|_F^2 \tag{30}$$

$$\leq \|x\|_F^2 \sqrt{2 - 2c} \tag{31}$$

$\square$

**Theorem 3.4.** *Let $H \in \mathbb{R}^{mn \times mn} = \nabla^2 \mathcal{L}(W^*)$ be the Hessian of a linear layer $W$ with respect to the KL to the original model outputs, $H_O \in \mathbb{R}^{m \times m}$ and $H_L \in \mathbb{R}^{n \times n}$ be two p.d. matrices, and $\mathcal{Q}$ perform nearest or stoch. rounding with $\mathbb{E}[(\mathcal{Q}(x) - x)^2] \leq \sigma^2$. Furthermore, let $W$ be the output of Equation 5 with $L_O', L_I'$ from the LDL decompositions of $H_O, H_I$, respectively. Then,*

$$\operatorname{vec}(\Delta) H \operatorname{vec}(\Delta)^T \leq \|H\|_F \Big( \|\Delta\|_F^2 \sqrt{2 - 2c} +$$

$$\frac{\mu_I^2 \mu_O^2}{mn \|H_I\|_F \|H_O\|_F} \operatorname{tr}(H_I^{1/2})^2 \operatorname{tr}(H_O^{1/2})^2 \sigma^2 \Big)$$

*where $c = \frac{\langle H, H_O \otimes H_I \rangle}{\|H\|_F \|H_O\|_F \|H_I\|_F}$ is the cosine similarity between $H$ and $H_O \otimes H_I$.*

*Proof.* From Lemma A.4, we have that

$$\left| \frac{\Delta W H \Delta W^T}{\|H\|} - \frac{\Delta W (H_O \otimes H_I) \Delta W^T}{\|H_O\|\|H_I\|} \right| \leq \|\Delta W\|_F^2 \sqrt{2 - 2c}. \tag{32}$$

Then,

$$\Delta W H \Delta W^T \leq \|H\| \left( \|\Delta W\|_F^2 \sqrt{2 - 2c} + \frac{\Delta W (H_O \otimes H_I) \Delta W^T}{\|H_O\|\|H_I\|} \right) \tag{33}$$

$$\Delta W H \Delta W^T \leq \|H\| \left( \|\Delta W\|_F^2 \sqrt{2 - 2c} + \frac{\mu_I^2 \mu_O^2}{mn\|H_I\|\|H_O\|} \operatorname{tr}(H_I^{1/2})^2 \operatorname{tr}(H_O^{1/2})^2 \sigma^2 \right). \tag{34}$$

$\square$

### A.8. Modified Pytorch Backward Pass to Compute Sketch A

```python
class LinearNoBias(torch.autograd.Function):
    @staticmethod
    @torch.amp.custom_fwd(device_type='cuda')
    def forward(ctx, input, weight, mode, parent_class):
        ctx.save_for_backward(input, weight)
        ctx.mode = mode
        ctx.parent_class = parent_class

        return input @ weight.T

    @staticmethod
    @torch.amp.custom_bwd(device_type='cuda')
    def backward(ctx, grad_output):
        it, reset, div = ctx.mode
        is_buffer = local_rank == ctx.parent_class.buffer_dev

        input, weight = ctx.saved_tensors
        ws = weight.shape
        grad_input = grad_output @ weight
        del weight

        if ctx.parent_class.collect_hess:
            grad_output = grad_output.reshape(-1, grad_output.shape[-1])
            input = input.reshape(-1, input.shape[-1])
            op_dtype = ctx.parent_class.op_dtype
            with torch.amp.autocast('cuda', enabled=False):
                grad_output = grad_output.float()
                input = input.float()
                bs = input.shape[0]
                if it == 0:
                    del grad_output
                    if reset and is_buffer:
                        ctx.parent_class.hin.mul_(0)

                    in_hess = sym_to_flat(input.T @ input) / ctx.parent_class.scale
                    del input
                    torch.distributed.reduce(in_hess, ctx.parent_class.buffer_dev, op=ReduceOp.AVG)

                    if is_buffer:
                        ctx.parent_class.hin.add_(in_hess.to(ctx.parent_class.hin.device).to(op_dtype))
                        ctx.parent_class.ct += bs / ctx.parent_class.scale
                        if div:
                            ctx.parent_class.hin.div_(ctx.parent_class.ct)
                            ctx.parent_class.ct = 0

                    del in_hess
                    torch.cuda.empty_cache()
                else:
                    if it % 2 == 0:
                        if reset and is_buffer:
                            ctx.parent_class.hin.mul_(0)

                        if not is_buffer:
                            out_hess = torch.zeros(ctx.parent_class.out_features * (ctx.parent_class.out_features +
    1)//2, dtype=op_dtype, device=local_rank)
                        else:
                            out_hess = ctx.parent_class.hout.to(local_rank)
```

```
                        torch.distributed.broadcast(out_hess, ctx.parent_class.buffer_dev)
                        out_hess = flat_to_sym(out_hess, ws[0]).float()
                        in_hess = input.T @ (input * ((grad_output @ out_hess) * grad_output).sum(dim=-1, keepdims=
    True)) / out_hess.norm()**2
                        del input, grad_output, out_hess
                        in_hess = sym_to_flat(in_hess) / ctx.parent_class.scale
                        torch.distributed.reduce(in_hess, ctx.parent_class.buffer_dev, op=ReduceOp.AVG)
                        if is_buffer:
                            ctx.parent_class.hin.add_(in_hess.to(ctx.parent_class.hin.device).to(op_dtype))
                            ctx.parent_class.ct += bs / ctx.parent_class.scale
                            if div:
                                ctx.parent_class.hin.div_(ctx.parent_class.ct)
                                ctx.parent_class.ct = 0

                        del in_hess
                    else:
                        if reset and is_buffer:
                            ctx.parent_class.hout.mul_(0)

                        if not is_buffer:
                            in_hess = torch.zeros(ctx.parent_class.in_features * (ctx.parent_class.in_features + 1)
    //2, dtype=op_dtype, device=local_rank)
                        else:
                            in_hess = ctx.parent_class.hin.to(local_rank)
                        torch.distributed.broadcast(in_hess, ctx.parent_class.buffer_dev)
                        in_hess = flat_to_sym(in_hess, ws[1]).float()
                        out_hess = grad_output.T @ (grad_output * ((input @ in_hess) * input).sum(dim=-1, keepdims=
    True)) / in_hess.norm()**2
                        del input, grad_output, in_hess
                        out_hess = sym_to_flat(out_hess) / ctx.parent_class.scale
                        torch.distributed.reduce(out_hess, ctx.parent_class.buffer_dev, op=ReduceOp.AVG)
                        if is_buffer:
                            ctx.parent_class.hout.add_(out_hess.to(ctx.parent_class.hout.device).to(op_dtype))
                            ctx.parent_class.ct += bs / ctx.parent_class.scale
                            if div:
                                ctx.parent_class.hout.div_(ctx.parent_class.ct)
                                ctx.parent_class.ct = 0

                        del out_hess

        torch.cuda.empty_cache()
        return grad_input.to(local_rank), None, None, None
```

## A.9. Modified Backward Pass to Compute Sketch B

```
class LinearNoBias(torch.autograd.Function):
    @staticmethod
    @torch.amp.custom_fwd(device_type='cuda')
    def forward(ctx, input, weight, mode, parent_class):
        ctx.save_for_backward(input, weight)
        ctx.mode = mode
        ctx.parent_class = parent_class

        return input @ weight.T

    @staticmethod
    @torch.amp.custom_bwd(device_type='cuda')
    def backward(ctx, grad_output):
        it, reset, div = ctx.mode
        is_buffer = local_rank == ctx.parent_class.buffer_dev

        input, weight = ctx.saved_tensors
        ws = weight.shape
        grad_input = grad_output @ weight
        del weight

        if ctx.parent_class.collect_hess:
            op_dtype = ctx.parent_class.op_dtype
            bs = input.shape[0]
            with torch.amp.autocast('cuda', enabled=False):
                if it == 0:
                    if reset and is_buffer:
                        ctx.parent_class.hin.mul_(0)

                    grad_output = grad_output.float()
                    input = input.float()
                    in_hess = sym_to_flat(torch.einsum('btm,btn,bsm,bsk->nk', grad_output, input, grad_output, input
    ))
                    handle_in = torch.distributed.reduce(in_hess, ctx.parent_class.buffer_dev, op=ReduceOp.AVG,
```

```
async_op=True)
                    out_hess = sym_to_flat(torch.einsum('btm,btn,bsk,bsn->mk', grad_output, input, grad_output,
input))
                    handle_out = torch.distributed.reduce(out_hess, ctx.parent_class.buffer_dev, op=ReduceOp.AVG,
async_op=True)
                    del grad_output, input
                    handle_in.wait()
                    handle_out.wait()

                    if is_buffer:
                        ctx.parent_class.hin.add_(in_hess.to(ctx.parent_class.hin.device).to(op_dtype))
                        ctx.parent_class.hout.add_(out_hess.to(ctx.parent_class.hout.device).to(op_dtype))
                        ctx.parent_class.ct += bs
                        if div:
                            ctx.parent_class.hin.div_(ctx.parent_class.ct)
                            ctx.parent_class.hout.div_(ctx.parent_class.ct)
                            ctx.parent_class.ct = 0

                    del in_hess, out_hess
                    torch.cuda.empty_cache()

        torch.cuda.empty_cache()
        return grad_input.to(local_rank), None, None, None
```

### A.10. Additional Results

Table 7 contains full zeroshot results from the "no finetuning" table in the main body. Table 8 shows results on Qwen 3 8B. Like with Llama and Gemma, YAQA outperforms LDLQ on Qwen.

Table 5 contains results comparing YAQA to CBQ, an older method that performs AdaRound-style rounding over a "sliding window" of decoder blocks. This allows it to capture cross-block dependencies beyond the immediate activation error. CBQ performs well, but is limited to scalar quantizers and is outperformed by YAQA . Another similar method, PV-Tuning, performs decoder block level quantization on the end-to-end error. PV-Tuning relies on learned vector quantizers, but is less effective on larger models. Furthermore, there are no PV-Tuned models with "standard" quantizers to compare against. However, LDLQ with a fixed trellis quantizer (i.e. QTIP) already outperforms PV-Tuning, so by association, so does YAQA.

Finally, Table 6 shows an experiment without incoherence processing. All quantized models in this table use an INT4 quantizer with a 16 bit groupwise absmax scale shared across 32 contiguous elements, giving an effective 4.5 bits per weight. We used $g_x = 1$ and $g_y = 32$ for all methods. Even with out incoherence processing, YAQA outperforms LDLQ.

*Table 5.* Results without finetuning on Llama 2 7B.

| ALGORITHM | INT2 | | INT3 | | INT4 | |
|---|---|---|---|---|---|---|
| | W2 | C4 | W2 | C4 | W2 | C4 |
| CBQ | 8.01 | 11.30 | 5.89 | 7.56 | **5.52** | 7.05 |
| YAQA-B | **7.45** | **9.22** | **5.82** | **7.38** | 5.54 | **7.04** |

*Table 6.* Results without incoherence processing. All results are without finetuning and use an INT4 quantizer with a 16-bit groupwise scale shared across 32 contiguous elements (4.5 bits).

| ALGO. | $D_{KL} \downarrow$ | PPL $\downarrow$ | | 0-SHOT ACC $\uparrow$ | | | | | |
|---|---|---|---|---|---|---|---|---|---|
| | W2 | W2 | C4 | AVG | ARCC | ARCE | BOOLQ | HSWAG | PIQA |
| BF16 | 0 | 6.50 | 8.02 | 69.82 | 51.37 | 78.03 | 82.05 | 57.74 | 79.92 |
| LDLQ | 0.033 | 6.75 | 8.21 | 68.35 | 49.74 | 77.36 | 78.35 | 56.83 | 79.49 |
| **YAQA-A** | 0.025 | 6.67 | 8.17 | **69.36** | 49.74 | 77.65 | 81.62 | 57.16 | 80.63 |
| **YAQA-B** | **0.021** | **6.65** | **8.15** | 68.95 | 50.68 | 78.20 | 79.72 | 57.06 | 79.11 |

### A.11. Number of Sequences Needed for Sketch B

Table 9 measures the effect of the number of sequences used for Sketch B. Although we used 64K sequences for our main results, using as few as 2K sequences, which is even cheaper than LDLQ, results in state-of-the-art KL divergence and downstream performance.

*Table 7.* Full zeroshot accuracy results for Table 1. Higher is better.

| MODEL | ALGO. | QUANT. | 0-SHOT ACC ↑ | | | | |
|---|---|---|---|---|---|---|---|
| | | | ARCC | ARCE | BOOLQ | HSWAG | PIQA |
| 3.1 70B INST | BF16 | | 56.40 | 75.34 | 62.20 | 61.51 | 82.92 |
| | LDLQ | QTIP 2 | 52.05 | 73.91 | 62.17 | 58.10 | 81.01 |
| | | QTIP 3 | 56.06 | 75.88 | 62.23 | 61.00 | 82.70 |
| | | QTIP 4 | 55.72 | 75.63 | 62.29 | 61.18 | 83.08 |
| | YAQA-A | QTIP 2 | 52.82 | 74.07 | 62.17 | 58.27 | 82.26 |
| | | QTIP 3 | 56.07 | 75.88 | 62.20 | 61.84 | 82.46 |
| | | QTIP 4 | 56.06 | 75.42 | 62.17 | 61.11 | 82.75 |
| | YAQA-B | QTIP 2 | 54.44 | 73.44 | 62.17 | 59.24 | 81.66 |
| | | QTIP 3 | 55.72 | 74.33 | 62.17 | 60.91 | 83.08 |
| | | QTIP 4 | 56.31 | 76.09 | 62.29 | 61.10 | 82.86 |
| 3.1 8B INST | BF16 | | 51.37 | 78.03 | 82.05 | 57.74 | 79.92 |
| | LDLQ | QTIP 2 | 41.89 | 74.28 | 67.98 | 51.67 | 76.71 |
| | | QTIP 3 | 50.34 | 77.61 | 82.05 | 56.57 | 79.87 |
| | | QTIP 4 | 50.68 | 78.07 | 80.98 | 57.32 | 79.98 |
| | YAQA-A | QTIP 2 | 45.39 | 73.91 | 70.34 | 52.59 | 78.07 |
| | | QTIP 3 | 49.83 | 77.23 | 81.76 | 56.85 | 79.76 |
| | | QTIP 4 | 50.34 | 78.32 | 82.45 | 57.35 | 79.65 |
| | YAQA-B | QTIP 2 | 44.20 | 75.08 | 70.64 | 52.91 | 78.78 |
| | | QTIP 3 | 51.02 | 77.86 | 81.04 | 57.24 | 79.38 |
| | | QTIP 4 | 50.94 | 78.49 | 82.02 | 57.47 | 79.98 |
| 3.2 3B INST | BF16 | | 44.45 | 71.42 | 76.18 | 51.19 | 75.73 |
| | LDLQ | QTIP 2 | 32.00 | 58.50 | 69.42 | 45.57 | 72.91 |
| | | QTIP 3 | 41.72 | 70.66 | 74.19 | 50.46 | 75.24 |
| | | QTIP 4 | 43.52 | 71.80 | 75.93 | 51.03 | 76.22 |
| | YAQA-A | QTIP 2 | 38.65 | 66.46 | 66.76 | 45.98 | 73.61 |
| | | QTIP 3 | 42.15 | 69.70 | 74.92 | 50.04 | 75.73 |
| | | QTIP 4 | 42.66 | 70.24 | 76.57 | 50.85 | 76.06 |
| | YAQA-B | QTIP 2 | 38.05 | 68.22 | 68.75 | 47.11 | 73.39 |
| | | QTIP 3 | 42.75 | 69.95 | 75.66 | 50.38 | 74.97 |
| | | QTIP 4 | 43.60 | 71.04 | 75.96 | 50.74 | 75.19 |
| 3.2 1B INST | BF16 | | 32.85 | 57.24 | 66.09 | 44.74 | 73.01 |
| | LDLQ | QTIP 2 | 27.30 | 51.60 | 61.71 | 38.66 | 68.72 |
| | | QTIP 3 | 29.44 | 54.42 | 65.47 | 43.31 | 71.82 |
| | | QTIP 4 | 30.46 | 54.76 | 65.84 | 43.70 | 72.58 |
| | YAQA-A | QTIP 2 | 27.59 | 51.55 | 62.84 | 39.28 | 68.86 |
| | | QTIP 3 | 32.17 | 56.02 | 65.47 | 43.51 | 72.14 |
| | | QTIP 4 | 32.51 | 56.86 | 64.37 | 44.33 | 72.80 |
| | YAQA-B | QTIP 2 | 27.47 | 54.46 | 62.51 | 39.89 | 70.18 |
| | | QTIP 3 | 31.31 | 56.44 | 62.60 | 43.20 | 73.50 |
| | | QTIP 4 | 31.48 | 55.18 | 64.86 | 43.42 | 74.27 |

*Table 8.* Results on Qwen 3 8B with the QTIP quantizer, incoherence processing, and no finetuning. Like on Llama and Gemma models, YAQA achieve state-of-the-art performance on Qwen models.

| QWEN3 8B | BITS | W2 PPL (4K CTX) | W2 KL (4K CTX) | MMLU 0 SHOT | GSM8K 4 SHOT COT EXACT MATCH |
|---|---|---|---|---|---|
| UNQUANTIZED | 16 | 9.00 | 0 | 72.99 | 84.31 |
| LDLQ | 2 | 10.81 | 0.285 | 63.94 | 56.31 |
| YAQA-B | 2 | 10.16 | 0.205 | 66.75 | 66.07 |
| LDLQ | 3 | 9.31 | 0.0681 | 70.40 | 80.81 |
| YAQA-B | 3 | 9.20 | 0.0479 | 71.36 | 82.55 |
| LDLQ | 4 | 9.07 | 0.0216 | 73.04 | 82.38 |
| YAQA-B | 4 | 9.04 | 0.0153 | 72.78 | 83.81 |

*Table 9.* Results quantizing Qwen 3 8B to 2 bits with the QTIP quantizer, incoherence processing, and no finetuning. Sketch B is robust to the number of sequences used for Hessian sketching.

| QWEN3 8B | SEQS. | GPU-HRS | BITS | W2 PPL (4K CTX) | W2 KL (4K CTX) | MMLU 0 SHOT | GSM8K 4 SHOT COT EXACT MATCH |
|---|---|---|---|---|---|---|---|
| ORIGINAL | | | 16 | 9.00 | 0 | 72.99 | 84.31 |
| LDLQ | 4K | 1.5 | 2 | 10.81 | 0.285 | 63.94 | 56.31 |
| YAQA-B | 2K | 1 | 2 | 10.33 | 0.227 | 65.74 | 62.14 |
| YAQA-B | 4K | 2 | 2 | 10.32 | 0.221 | 66.04 | 63.82 |
| YAQA-B | 8K | 4 | 2 | 10.34 | 0.212 | 66.65 | 63.34 |
| YAQA-B | 16K | 7 | 2 | 10.15 | 0.210 | 66.29 | 64.86 |
| YAQA-B | 32K | 15 | 2 | 10.20 | 0.206 | 66.59 | 67.02 |
| YAQA-B | 64K | 30 | 2 | 10.16 | 0.205 | 66.75 | 66.07 |

