# OpenReview forum: "Model-Preserving Adaptive Rounding"
_ICML.cc/2026/Conference — ICML 2026 regular_

### Official Review · Reviewer_eCGv · 2026-03-09

**Soundness:** 2
**Presentation:** 3
**Significance:** 2
**Originality:** 3
**Overall Recommendation:** 4
**Confidence:** 4

**Summary:**

The authors proposes a new adaptive rounding method named Yet Another Quantization Algorithm (YAQA), which can directly consider the error at the network’s output and thus optimize the end-to-end KL. They conduct experiments on Llama 3.1 and 3.2 models with different bit-width configurations.

**Compliance With Llm Reviewing Policy:**

Affirmed.

**Final Justification:**

Most of my concerns have been addressed, but my question about reasoning models or MoE models still remains unresolved, because the authors only mention that "there is no reason why YAQA could not be used for these models." Accordingly, I finally lean toward "Weak Accept".

**Key Questions For Authors:**

Is YAQA also effective for reasoning models or MoE models?

**Strengths And Weaknesses:**

**Strengths**

- They provide theoretical bounds on the end-to-end quantization error.

- They show that YAQA can reduces the KL divergence compared to existing quantization techniques.

- They presents the experimental results of YAQA across various bit-width settings.

**Weaknesses**

- Either PPL or CSR accuracy is reported. So, it is hard to determine whether YAQA is effective or not. It would be more instructive if the authors also explore more challenging tasks (e.g., MMLU and/or its variants) and generation tasks (e.g., IFEval, GSM8K). Considering that the authors highlight that YAQA reduces the KL divergence, it seems necessary to compare YAQA with existing quantization algorithms on generation benchmarks.

- They focus on only one model family (i.e., Llama). To further validate the efficacy of YAQA, it would be more beneficial if they also conduct experiments on Qwen2.5 or Qwen3.

---

> ### Author Rebuttal · Authors · 2026-03-30
>
> **Dear Reviewer, thank you for your review. Due to space constraints, we have omitted the header we included in our responses to RQ7C and UAH1. We request that you read this header since it also applies to your review. Again due to the character limit, the second half of this response is a continuation of our response to gyJv. It is not related to your review.**
>
> **Response to eCGv:**
>
> - Qwen3, MMLU, GSM8K
>   - As requested, we ran experiments on Qwen 3 8B in addition to the Llama and Gemma results already in the paper. We also reported MMLU and GSM8K numbers for the Qwen experiments. At 2 bits, YAQA performs significantly better than LDLQ (+2-3% on MMLU, +5-7% on GSM8K). At 3 and 4 bits, YAQA is close to lossless and still consistently improves upon LDLQ. These results are state of the art, in line with what we reported in our original submission, and way better than any of the baselines presented in arXiv:2505.02214. The numbers below use the QTIP quantizer, incoherence processing, and no finetuning.
> | Qwen3 8B | Bits | W2 PPL (4k ctx) | W2 KL (4k ctx) | MMLU 0 shot | GSM8K 4 shot COT Exact Match |
> |:---:|:---:|:---:|:---:|:---:|:---:|
> | Unquantized | 16 | 9.00 | 0 | 72.99 | 84.31 |
> | LDLQ | 2 | 10.81 | 0.285 | 63.94 | 56.31 |
> | **YAQA-B** | **2** | **10.16** | **0.205** | **66.75** | **66.07** |
> | LDLQ | 3 | 9.31 | 0.0681 | 70.40 | 80.81 |
> | **YAQA-B** | **3** | **9.20** | **0.0479** | **71.36** | **82.55** |
> | LDLQ | 4 | 9.07 | 0.0216 | **73.04** | 82.38 |
> | **YAQA-B** | **4** | **9.04** | **0.0153** | 72.78 | **83.81** |
>   - You may also be interested in experiments we ran using different devset sizes. These results are in our response to UAH1. Although we used 64K sequences in the paper, **using as few as 2K sequences, which is even cheaper than LDLQ, still results in state-of-the-art KL divergence and downstream results.** At all devset sizes, YAQA significantly outperforms LDLQ.
> - Reasoning models and MoEs: there is no reason why YAQA could not be used for these models.
>
> **Continuation of response to gyJv**:
>
> - YAQA’s empirical advantage
>   - The premise of YAQA is that there are much better Hessians for adaptive rounding than Adaround’s. The question is what makes a good Hessian and how to obtain it, which is what we answer. The main difference between YAQA and LDLQ is the Hessian used. We’re not sure what you mean by broader optimization setup. LDLQ is running adaptive rounding on $H_1$, YAQA is running adaptive rounding on $H_O \otimes H_I$. Eq. 6 is just a rewriting of the operations being performed. We’re also not sure what you mean by “additional compute budget.” There is no explicit compute budget that limits LDLQ. If you were referring to the cost of Hessian computation itself (i.e. “calibration”), our response to UAH1 shows that this cost can be significantly lower (in fact, even lower than LDLQ) without degrading quality.
> - Gemma QAT
>   - First, we’d like to make an important distinction about QAT. Frontier labs have largely converged on QAT as a form of continued pretraining where the last part of training is done in low precision. This is not the same as distillation to the original model, or “QAD.”
>   - We chose Gemma’s QAT model for many reasons. First, at the time of writing, it was the only open source model where both the pre and post QAT BF16 weights were available. Second, the QAT process was performed by Google, who trained the model. This means they had access to the original training data, much more compute than any academic paper, and are generally quite competent. In contrast, if we had chosen some other academic QAT work, none of this would have been guaranteed, especially the data, which is very important for training. Older works such as LSQ (2019) have no official LLM implementations, which means we would have to implement a baseline unfamiliar to ourselves. Furthermore, papers such as arXiv:2406.06385v3 and 2402.04396 have shown that LSQ does not even consistently outperform GPTQ and LDLQ.
>   - The point of the QAT comparison was just to show that QAT as commonly done today is actually pretty bad at preserving the original model. **We are not claiming that QAT is useless or bad in any other way**. We’d be more than happy to rephrase our QAT comparison to emphasize this if you’d like.
> - Theory vs design
>   - To reiterate, the theory in YAQA directly informs and supports the practical design choices we made. The SND tells you exactly what forms of Hessians are tractable. We chose a Kronecker factored one because the SND is low (Lemma 3.3), and there is lots of empirical evidence that LLM Hessians are $\approx$ low rank (arXiv:1706.04454, 2406.17748, 2307.13304). Our error bound tells us that we want to maximize the cosine similarity. This is exactly what our sketches are trying to do with power iteration, which provably converges to the rank-1 approximation that maximizes cosine similarity. Basically, every part YAQA's practical implementation was chosen based on the theory.

---

> > ### Author Rebuttal · Reviewer_eCGv · 2026-04-04
> >
> > Thank you for the detailed response. However, the experimental results the author presented during the rebuttal period seem unconvincing, because their experimental settings are not conventional. For example, the MMLU evaluation is generally based on 5-shot rather than 0-shot. In addition, using the exact match in the GSM8K evaluation is stringent. Accordingly, the FP results the authors presented on MMLU and GSM8K are 72.99 and 84.31, which are far less than the results of Qwen3 8B in the Qwen3 Technical Report (76.89 and 89.84 as shown in https://arxiv.org/pdf/2505.09388). As a result, I maintain my original score.

---

> > > ### Author Response · Authors · 2026-04-05
> > >
> > > We do not understand the goals of your response and review. In your original review, you asked about Qwen3 in addition to our existing experiments on Llama and Gemma. You also explicitly stated that "it would be more instructive if the authors also explore more challenging tasks ... and generation tasks." This is exactly what we did in our response to you. We tested Qwen3 8B Base on MMLU (0-shot) and GSM8K (4 CoT, exact match). In your follow up response, you stated that "the MMLU evaluation is generally based on 5-shot rather than 0-shot. In addition, using the exact match in the GSM8K evaluation is stringent." **Our presented evals were strictly more challenging than what you wanted to see and in line with your request for more challenging evals. Thus, we do not follow your logic in 1) asking for a more challenging benchmark in your original review and then 2) complaining in your second response that the benchmark we chose was too challenging**.
> > >
> > > Regardless, we have rerun our benchmarks using your desired settings: MMLU 5-shot and GSM8K 4-shot CoT flexible extract. As a reminder, for both LDLQ and YAQA, we used the QTIP quantizer, incoherence processing with the random Hadamard transform (RHT), and no recovery finetuning. The Qwen3 tech report did not state the exact settings or codebase that they used for each, only stating that they used "MMLU 5-shot" and "GSM8K 4-shot CoT." We used lm-eval, as is standard in quantization papers (see QuIP#, QTIP, AQLM, PV-Tuning, etc.). Multiple people have reported that they were unable to reproduce the official Qwen tech report numbers with lm-eval (e.g. https://github.com/EleutherAI/lm-evaluation-harness/issues/3129). However, without the official Qwen MMLU and GSM8K eval codebase, it is impossible to do better than lm-eval. For GSM8K, we ran both the `gsm8k` and `gsm8k_cot` configs in lm-eval since again, Qwen did not state what their exact config was.
> > >
> > >
> > > MMLU 5 shot
> > > - BF16: 74.97
> > > - LDLQ 2 bit: 66.89
> > > - YAQA 2 bit: 68.95
> > > - LDLQ 3 bit: 73.11
> > > - YAQA 3 bit: 73.90
> > > - LDLQ 4 bit: 74.35
> > > - YAQA 4 bit: 74.70
> > >
> > > GSM8K 4 shot, CoT, Exact Match, Flexible Extract (gsm8k_cot, num_fewshot=4)
> > >
> > > - BF16 87.95
> > > - LDLQ 2 bit: 67.62
> > > - YAQA 2 bit: 75.89
> > > - LDLQ 3 bit: 85.44
> > > - YAQA 3 bit: 86.97
> > > - LDLQ 4 bit: 85.76
> > > - YAQA 4 bit: 87.34
> > >
> > > GSM8K 4 shot, CoT, Exact Match, Flexible Extract (gsm8k, num_fewshot=4)
> > >
> > > - BF16 88.10
> > > - LDLQ 2 bit: 71.24
> > > - YAQA 2 bit: 76.35
> > > - LDLQ 3 bit: 86.05
> > > - YAQA 3 bit: 86.42
> > > - LDLQ 4 bit: 87.35
> > > - YAQA 4 bit: 87.58
> > >
> > > **In all of these settings, including the more challenging settings in our original rebuttal, YAQA outperforms the previous state of the art method LDLQ.** This shows that the exact eval setting is irrelevant when determining which *quantization* method is better: YAQA or LDLQ. The purpose of this paper is to show that YAQA is a better adaptive rounding construction than LDLQ, and **all of our benchmarks show this.** Furthermore, Both LDLQ and YAQA outperform the original results we sent, which shows that indeed, our original setting was more challenging as you originally desired.
> > >
> > > **As further evidence that the quantization literature has not converged on a unified way to run Qwen 3 downstream evals, consider that ParoQuant (arxiv:2511.10645), D2Quant (2602.02546), "An Empirical Study of Qwen3 Quantization" (2505.02214) all report different MMLU numbers from each other and the Qwen3 tech report**. ParoQuant reports 74.6% for 8B instruct, D2Quant 72.96% for 8B instruct, the empirical study 76.7% for 8B base and 74.7% for 8B instruct, and the tech report 76.9% for 8B base and no number for 8B instruct. Clearly, none of these papers are in agreement with each other, but that does not matter for the purposes of evaluating quantization algorithms **within** a single paper, as long as the methodology is consistent.
> > >
> > > As a side note, both ParoQuant and D2Quant target pre-quantization weight and activation transformations (e.g. like the RHT), which are orthogonal to the adaptive rounding algorithm used. As such, it does not make sense to compare YAQA against them, since YAQA could be used with either one of them like we use YAQA with the RHT in our experiments. Furthermore, both ParoQuant and D2Quant came out within 2 months of the ICML submission deadline, qualifying them as concurrent work (not that it would make sense to compare to them, anyways).

---

### Official Review · Reviewer_gyJv · 2026-03-09

**Soundness:** 2
**Presentation:** 3
**Significance:** 3
**Originality:** 3
**Overall Recommendation:** 4
**Confidence:** 3

**Summary:**

This paper studies post-training quantization for LLMs and argues that standard layer-wise adaptive rounding methods optimize a poor proxy of the true end-to-end output error. To address this, the paper proposes YAQA, a model-preserving adaptive rounding algorithm that uses Kronecker-factored Hessian sketches to better approximate the full-model KL objective during quantization. The paper provides theoretical analysis relating adaptive rounding tractability to Hessian structure, derives an end-to-end error bound involving cosine similarity to the true Hessian, and introduces two practical Hessian sketching variants. Experiments suggest improved KL divergence and competitive downstream performance.

**Compliance With Llm Reviewing Policy:**

Affirmed.

**Final Justification:**

After considering both the paper and the rebuttal, I am updating my recommendation to weak accept.

**Key Questions For Authors:**

1. Can the authors broaden and better justify the choice of baselines in the empirical comparison? The current evaluation focuses mainly on LDLQ, GuidedQuant, DiscQuant, and one QAT comparison, while several widely recognized quantization methods are discussed but not directly evaluated.
2. How much of YAQA’s empirical advantage comes specifically from the proposed Hessian sketching, rather than from the broader optimization setup or additional compute budget?
3. Why should the Gemma QAT setup be viewed as the most appropriate QAT comparison here? Since the paper includes a QAT comparison but does not evaluate against older yet still widely used general-purpose QAT methods such as LSQ, I would like a more explicit justification for this choice.
4. Can the authors clarify how strongly the theoretical results support the specific practical design choices in YAQA? The paper derives an end-to-end error bound involving cosine similarity between the true Hessian and the Kronecker-factored approximation, and then uses this to motivate the sketch construction. However, it is still not entirely clear to me whether the theory identifies YAQA’s sketches as especially well justified, or mainly provides directional support for this class of approximations.

**Limitations:**

The authors should more clearly acknowledge the dependence on approximation choices, the restricted experimental scope, and the fact that the theoretical guarantees are still mediated through surrogate quantities rather than the exact full end-to-end objective.

**Strengths And Weaknesses:**

**Strengths**
1. The paper addresses an important problem for LLM quantization.
2. The method is technically coherent and reasonably well motivated.

**Weaknesses**
1. The novelty is weaker than the paper suggests. The manuscript presents YAQA as a major step forward, but much of the contribution appears to be a careful recombination of familiar ingredients: second-order layerwise quantization, Kronecker-factored curvature approximations, and sketching ideas related to Hessian approximations. The technical packaging is solid, but I am not fully convinced that the conceptual novelty is strong enough for ICML acceptance at its current level.
2. The theory is interesting but not as decisive as the paper’s framing implies. The end-to-end error bound is a positive aspect, but the analysis relies on several approximation layers and surrogate quantities, especially cosine similarity to the true Hessian and tractable sketches that are only indirectly connected to the exact end objective. As a result, the theory supports the method directionally, but does not fully establish that the proposed sketches are the uniquely right or most principled practical choice.

---

> ### Author Rebuttal · Authors · 2026-03-30
>
> **Dear reviewer, thank you for your review. Due to space constraints, we have omitted the header in our other rebuttals. We request you to read it as it applies to this rebuttal as well. Again due to the character limit, we have also split our response to you over this rebuttal and our rebuttal to eCGv. Please read that rebuttal for the rest of your questions.**
>
> - **Novelty and theory:**
>   - We respectfully disagree. There are 3 main contributions in YAQA, none of which have ever been proposed in the LLM quantization literature and collectively significantly advance the state of the field. **YAQA includes novel theoretical insights on the tractability of adaptive rounding (a problem that no other paper has studied), the first end-to-end error bounds on quantization error (which no other paper has given), and state of the art empirical results.**
>   - First, in S3.1.1, we characterize when adaptive rounding is tractable via the structure of the Hessian approximation (Def. 3.1 and Lemma 3.2). Using the structural nilpotence degree (SND), a novel construct in itself, we show that a Kronecker-factored Hessian approximation is both tractable while fitting the $\approx$ low-rank nature of LLM Hessians. Prior to YAQA, almost every adaptive rounding work Adaround Hessian $H_1 \ \mathbb{E}[xx^T]$ without much justification, which itself was initially proposed as an ad-hoc approximation. With the SND, we showed that there were actually a wide range of Hessian structures that admitted tractable adaptive rounding, such as our Kronecker-factored approximation and GuidedQuant’s block-diagonal approximation. **No other work before us had attempted to characterize what classes of Hessian sketches admitted tractable adaptive rounding, meaning that prior works were essentially shooting in the dark.**
>   - **Second, in Thm. 3.4, we gave the first end-to-end error bound for any LLM quantization algorithm. The strongest prior bounds only covered the immediate layerwise activation error (arXiv 2307.13304, 2507.18553, 2601.17187).** Our bound shows why we want to maximize the cosine similarity between the sketch and true Hessian. Similar to the structure of the Hessian, before us, people had largely chosen arbitrary Hessian estimates. $H_1$ is the Hessian of the immediate layerwise error, but it is actually not even optimal within its structure class. Likewise, GuidedQuant's Hessian is just a different $H_1$ for every channel group, which again is not the optimal block diagonal approximation. Finally, in our empirical evaluations, YAQA showed state-of-the-art performance, beating all prior PTQ methods even beyond adaptive rounding. YAQA even achieved a lower KL than Google’s official QAT of Gemma 3 12B, a model that Google themselves trained. Ignoring our theory, this would already be a major step forward in LLM quantization.
>   - Like you stated above, it wasn’t clear that prior Hessians were “uniquely right or [the] most principled practical choice.” In YAQA, we showed that they weren’t and demystified adaptive rounding by clearly defining (1) what structured Hessians admit fast adaptive rounding and (2) what properties the Hessian sketch should have. **We want to emphasize just how little the literature has understood adaptive rounding.** Without (1), it isn’t even possible to know where to begin estimating a Hessian. Without (2), we end up with methods like LDLQ and GuidedQuant that largely just guess at what the sketch should be. The fact that our empirical results are so much better shows why our contributions in (1) and (2) are important.
>   - With respect to your specific comments, we are not sure what you mean by "the analysis relies on ... approximation layers and surrogate[s]” We used the 2nd order approximation of the KL, which is standard in quantization and optimization, and the trace-incoherence bound from QuIP. Thm. 3.4 would give the same conclusions if we directly used $tr(D)$ instead. Since you generally cannot get a tractable 3rd order approximation of the KL, the former is basically as tight as you can get. **We are not claiming that our bound is the tightest you can achieve, but remember that prior to this bound, there were no bounds at all on the E2E error in the literature. Going from 0 to something useful is still, in our view, pretty impactful.**
>
> - **Baseline Justification**:
>   - We put a pretty lengthy description of our reasoning on pg. 6 already. To reiterate, almost every state of the art post training quantization paper in the literature uses GPTQ or LDLQ as for rounding. The entire QuIP series, NestQuant, and many other papers use LDLQ. QuaRot, SpinQuant, and the countless rotation matrix papers that do weight-activation quantization use GPTQ for weights. It is widely accepted that these papers all outperform older ones like OmniQuant, SmoothQuant, & AWQ. LDLQ-based methods like QTIP also outperform methods like AQLM and PV-Tuning, so comparing them would simply be repeating the literature.

---

> > ### Author Rebuttal · Reviewer_gyJv · 2026-04-03
> >
> > Thank you for the detailed rebuttal. It addresses my main concerns, and I will raise my score to weak accept.

---

### Official Review · Reviewer_UAH1 · 2026-03-11

**Soundness:** 3
**Presentation:** 3
**Significance:** 3
**Originality:** 3
**Overall Recommendation:** 5
**Confidence:** 3

**Summary:**

This study proposes Yet Another Quantization Algorithm (YAQA), a novel rounding algorithm which takes into account end-to-end error. Starting from rounding algorithm of LDLQ, authors find out conditions for hessian sketch to become tractable. Moivated on the analysis, a form Kronecker-factored Hessian sketch is proposed. Two versions are suggested for practical implementation, one assuming tokens are independent within a layer and another running a single round of poser iteration. Comparison with SOTA rounding methods on quantization of LLaMa and Gemma is given as experimental results

**Compliance With Llm Reviewing Policy:**

Affirmed.

**Final Justification:**

The authors’ rebuttal has addressed all of my concerns rasied on the review. Therefore, I am increasing my score

**Key Questions For Authors:**

+ Could the authors evaluate YAQA-B with different numbers of sequences and report the corresponding runtime?
+ It would be helpful if the authors could include a comparison of memory consumption and latency with other methods

**Limitations:**

While the possibility of severe negative impact or limitations is minimal, they are not specifically addressed in the paper

**Strengths And Weaknesses:**

[Strengths]
+ Provide valuable insight about tractable hessian sketch and mathematically showed bound of end-to-end error.
+ Provide two practical implementations of hessian sketch.
+ Consistent experimental results, surpassing other baseline methods in most cases even when recovery finetuning step is included.
+ Paper is well-organized, easy to follow

[Weaknesses]
+ While two variants of hessian sketch are proposed for practical implementations, the computational overhead induced by these calculation still appears significant.
+ Reagrding sketch B, it would be useful to evaluate whether reducing the number of token sequences can decrease latency while maintaining performance. Although normalized cosine similairy is reported, the paper does not provide evaluation results or runtime anlysis with different numbers of sequences.
+ Comparison of memory consumption and latency with other PTQ methods would be helpful, but is not provided in the paper

---

> ### Author Rebuttal · Authors · 2026-03-30
>
> Dear reviewer, thank you for your review. As you and others noted, YAQA “addresses an important problem for LLM quantization” (gyJv) by “provid[ing] valuable insight[s] about” the tractability of adaptive rounding and its end-to-end error (UAH1, RQ7C, eCGv). YAQA consists of three main components. First, we provide theory on what structural classes of Hessian estimates admit tractable adaptive rounding. Then, to the best of our knowledge, we provide the first end-to-end error bound for any LLM quantization algorithm and show that the optimal Hessian sketch within a structural class is one that maximizes cosine similarity. This results in two “practical implementations” of Hessian sketches that consistently “perform better than [baselines]” (UAH1, RQ7C). Finally, we provide an empirical evaluation of YAQA and show that it achieves state of the art results. YAQA outperforms existing baselines spanning a wide range of methods, including QAT, and is “amenable to recovery finetuning” (RQ7C, UAH1), an important part of modern quantization pipelines. Most importantly, YAQA consistently “reduces the KL divergence” over baselines, which is what we set out to do.
>
> As requested, we ran an experiment using different numbers of sequences for sketch B. Below are results quantizing Qwen 3 8B to 2 bits with the QTIP quantizer, incoherence processing, and no finetuning. YAQA is robust to the number of sequences used for Hessian sketching. Although we used 64K sequences in the paper, **using as few as 2K sequences, which is even cheaper than LDLQ, results in state-of-the-art KL divergence and downstream results.** At all devset sizes, YAQA significantly outperforms LDLQ. We believe this answers your first two questions, especially regarding the cost of calibration.
>
> | Qwen3 8B | Seqs | Hessian GPU-Hrs | Bits | W2 PPL (4k ctx) | W2 KL (4k ctx) | MMLU 0 shot | GSM8K 4 shot COT Exact Match |
> |:---:|:---:|:---:|:---:|:---:|:---:|:---:|:---:|
> | Original |  |  | 16 | 9.00 | 0 | 72.99 | 84.31 |
> | LDLQ | 4K | 1.5 | 2 | 10.81 | 0.285 | 63.94 | 56.31 |
> | YAQA-B | 2K | 1 | 2 | 10.33 | 0.227 | 65.74 | 62.14 |
> | YAQA-B | 4K | 2 | 2 | 10.32 | 0.221 | 66.04 | 63.82 |
> | YAQA-B | 8K | 4 | 2 | 10.34 | 0.212 | 66.65 | 63.34 |
> | YAQA-B | 16K | 7 | 2 | 10.15 | 0.210 | 66.29 | 64.86 |
> | YAQA-B | 32K | 15 | 2 | 10.20 | 0.206 | 66.59 | 67.02 |
> | YAQA-B | 64K | 30 | 2 | 10.16 | 0.205 | 66.75 | 66.07 |
>
> With regards to your last question on memory consumption and latency, the premise of YAQA is that it is possible to obtain a significantly better quantized model by spending slightly more time on adaptive rounding via a more informative Hessian estimate, all while still being tractable. The theory in this paper about the structural degree of the Hessian approximation directly characterizes what structured Hessian approximations admit tractable adaptive rounding, and the rest of the paper (error bounds, Hessian sketches) shows how to make this work on actual LLMs. By construction, the adaptive rounding part of YAQA is intended to be more expensive than LDLQ, while still being cheap to run.
>
> Running YAQA with INT4 on a 16K x 16K matrix takes well under a minute on a modern GPU (2.6s on a B200, 15.5s on a RTX 6000 Ada, 26s on an A6000). In fact, even on the legendary 1080Ti that is now 9 years old and has no tensor cores, it still only takes 2.5 minutes. On a modern GPU, adaptive rounding with YAQA and INT4 on an 8B model takes well under 1 GPU hour. The GPU memory needed for the adaptive rounding step is bottlenecked by taking the Cholesky of HO and HI. Since the largest dimension of modern LLMs is << 10^5, this is doable on a single GPU. This is also the same GPU memory cost as LDLQ and GPTQ. There, they only do HI, but HO and HI are essentially the same size and can be processed serially in YAQA, making the bottleneck the same.
>
> If we take these two components together (Hessian sketching and adaptive rounding), the cost of YAQA is a rounding error compared to training the model and the inference lifetime of the model. Even compared to only training the model, YAQA costs <<0.01% of the data and compute. The point of quantization is to get inference savings, and practically, the people quantizing models are the ones training them and doing large scale deployment. Quantized models reap savings every time they are run (i.e. over the entire inference lifespan of the model), meaning that it basically always makes sense to get the highest quality quantized model you can. This is why frontier labs and companies perform quantization aware training (QAT) for billions of tokens before deployment, even though this is orders of magnitude more expensive than YAQA or any other PTQ algorithm.

---

> > ### Author Rebuttal · Reviewer_UAH1 · 2026-04-03
> >
> > Thank you for the thorough and clarifying rebuttal. As my concerns have been satisfactorily resolved, I will increase my score.

---

### Official Review · Reviewer_RQ7C · 2026-03-13

**Soundness:** 3
**Presentation:** 4
**Significance:** 3
**Originality:** 4
**Overall Recommendation:** 5
**Confidence:** 4

**Summary:**

This paper introduces Yet another quantization algorithm for post-training quantization of LLMs. They generalize the local adaptive rounding methods by using end to end KL divergence of the quantized model outputs. They approximate the Hession with a Kronecker decomposition and propose two sketches to estimate the Kronecker cores. They perform a thorough theoretical and empirical analysis of their method, showing the advantages of their method.

**Compliance With Llm Reviewing Policy:**

Affirmed.

**Final Justification:**

The method described in the paper is theoretically well motivated, and the experiments cover a wide range of models and sizes. For these reasons, I assigned an accept score.

**Key Questions For Authors:**

1. The authors provide the time taken for the calibration phase for a 10B model. How does this scale across model sizes like the 3B and 70B parameter model
2. In the paper, they use RedPajama for all the Hessians. How do the different calibration datasets affect the performance of the model?
3. What is the wall clock time for quantizing with YAQA? How much GPU memory does it need for different model sizes?

**Limitations:**

yes

**Strengths And Weaknesses:**

**Strengths**
- They provide both theoretical and empirical analysis of their method on the end-to-end performance of the model.
- The two sketches for the approximate Hessian trade bias vs speed, while both methods generally perform better than the baseline methods.
- YAQA is amenable to recovery finetuning and achieves the best results in this setting as well

**Weaknesses**
- As the authors have mentioned, YAQA takes twice the time compared to LDLQ because of the output side feedback.
- To capture sequence mixing accurately, sketch B requires a large calibration set and nearly 30 GPU hours for a 10B model
- Even with higher costs, YAQA B underperforms YAQA A in downstream performance.

---

> ### Author Rebuttal · Authors · 2026-03-30
>
> Dear reviewer, thank you for your review. As you and others noted, YAQA “addresses an important problem for LLM quantization” (gyJv) by “provid[ing] valuable insight[s] about” the tractability of adaptive rounding and its end-to-end error (UAH1, RQ7C, eCGv). YAQA consists of 3 main components. First, we provide theory on what structural classes of Hessian estimates admit tractable adaptive rounding. Then, we provide the first end-to-end error bound for any LLM quantization algorithm and show that the optimal Hessian sketch is one that maximizes cosine similarity. This results in two “practical implementations” of Hessian sketches that consistently “perform better than [baselines]” (UAH1, RQ7C). Finally, we provide an empirical evaluation of YAQA and show that it achieves state of the art results. YAQA outperforms existing baselines spanning a wide range of methods, including QAT, and is “amenable to recovery finetuning” (RQ7C, UAH1), an important part of modern quantization pipelines. Most importantly, YAQA consistently reduces the KL divergence over baselines, which is what we set out to do.
>
> Below, we have responded to your individual questions.
>
> - Sketch B requires a large devset …
>   - In our response to UAH1, we included experiments on Qwen3 8B with different calibration set sizes; we refer you to that response for exact numbers. YAQA is robust to the number of sequences used for Hessian sketching. Although we used 64K sequences in the paper, **using as few as 2K sequences, which is even cheaper than LDLQ, results in state-of-the-art KL divergence and downstream results.** At all devset sizes, YAQA significantly outperforms LDLQ.
> - B vs A downstream perf.
>   - We did not consistently observe B to be worse than A, and in most cases, B is better than A. However, downstream 0-shot accuracy is known to be noisy. In fact, some of our quantized models outperform the 16 bit baselines.
> - Hessian sketch cost scaling (3B & 70B)
>   - The Hessian sketching cost is roughly linear in the # of FLOPs in the model. Using the O(N) FLOPs estimate rule where the # of FLOPs is linear in the # of parameters, we expect Sketch B to take 10-200 GPU hours for a 70B model depending on the # of sequences used (see UAH1). 200 GPU hours is still only one node for one day (far below the cost of training and inference), and using 65K sequences is probably overkill as our experiments above suggest.
> - RedPajama
>   - We used RedPajama because that is what prior works like QuIP# and QTIP use. We have not studied the effect of the calibration dataset on quantization, but some other works such as https://arxiv.org/abs/2311.09755 have. One option that is actually data free is to use unbiased rollouts from the model as calibration sequences, which we leave for future work.
> - Quantization step wall clock time and memory for YAQA vs LDLQ (first weakness & last question)
>   - On the actual quantization side, the premise of YAQA is that it is possible to obtain a significantly better quantized model by spending slightly more time on adaptive rounding via a more informative Hessian estimate, all while still being tractable. The theory in this paper about the structural degree of the Hessian approximation directly characterizes what structured Hessian approximations admit tractable adaptive rounding, and the rest of the paper (error bounds, Hessian sketches) shows how to make this work on actual LLMs. By construction, the adaptive rounding part of YAQA is intended to be more expensive than LDLQ, while still being cheap to run.
>   - Running YAQA with INT4 on a 16K x 16K matrix takes well under a minute on a modern GPU (2.6s on a B200, 15.5s on a RTX 6000 Ada, 26s on an A6000). In fact, even if we use the 9 year old 1080Ti that has no tensor cores, it still only takes 2.5 minutes. On a modern GPU, adaptive rounding with YAQA and INT4 on an 8B model takes well under 1 GPU hour. The GPU memory needed for the adaptive rounding step is bottlenecked by taking the Cholesky of HO and HI. Since the largest dimension of modern LLMs is << 10^5, this is doable on a single GPU. This is also the same GPU memory cost as LDLQ and GPTQ. There, they only do HI, but HO and HI are essentially the same size and can be processed serially in YAQA, making the bottleneck the same.
>
> More generally, regarding the cost of YAQA, if we consider both Hessian sketching and adaptive rounding together, the cost of YAQA is still a rounding error compared to training the model and the inference lifetime of the model. Even compared to only training the model, YAQA costs <<0.01% of the data and compute. The point of post-training quantization is to get inference savings, and practically, the people quantizing models are the ones training them and doing large scale deployment. Quantized models reap savings every time they are run (i.e. over the entire inference lifespan of the model), meaning that it basically always makes sense to get the highest quality quantized model you can.

---

> > ### Author Rebuttal · Reviewer_RQ7C · 2026-04-04
> >
> > The rebuttal answers all my questions, and I am keeping my favorable score.

---

### Decision · Program_Chairs · 2026-04-30

**Decision:**

Accept (regular)

**Comment:**

This paper proposes a new end-to-end network quantization algorithm for post-training quantization. While most existing methods focus on minimizing local errors introduced by quantization, the authors instead aim to preserve the KL divergence at the final layer. The resulting objective is decomposed into layer-wise subproblems using Hessian information, and the authors identify conditions under which the Hessian can be effectively approximated. Empirical results demonstrate that the proposed method outperforms baseline approaches, and the reviewers reached a clear consensus that the paper makes a sufficient contribution for acceptance.

If accepted, the authors should incorporate additional experiments and clarifications discussed in their response into the final version of the manuscript.